# Skin vasculature and hair follicle cross-talking associated with stem cell activation and tissue homeostasis

Kefei Nina Li[†], Prachi Jain[†], Catherine Hua He, Flora Chae Eun, Sangjo Kang, Tudorita Tumbar*

Molecular Biology and Genetics, Cornell University, Ithaca, United States

**Abstract** Skin vasculature cross-talking with hair follicle stem cells (HFSCs) is poorly understood. Skin vasculature undergoes dramatic remodeling during adult mouse hair cycle. Specifically, a horizontal plexus under the secondary hair germ (HPuHG) transiently neighbors the HFSC activation zone during the quiescence phase (telogen). Increased density of HPuHG can be induced by reciprocal mutations in the epithelium (*Runx1*) and endothelium (*Alk1*) in adult mice, and is accompanied by prolonged HFSC quiescence and by delayed entry and progression into the hair growth phase (anagen). Suggestively, skin vasculature produces BMP4, a well-established HFSC quiescence-inducing factor, thus contributing to a proliferation-inhibitory environment near the HFSC. Conversely, the HFSC activator Runx1 regulates secreted proteins with previously demonstrated roles in vasculature remodeling. We suggest a working model in which coordinated remodeling and molecular cross-talking of the adult epithelial and endothelial skin compartments modulate timing of HFSC activation from quiescence for proper tissue homeostasis of adult skin.
DOI: https://doi.org/10.7554/eLife.45977.001

**\*For correspondence:**
tt252@cornell.edu

[†]These authors contributed equally to this work

**Competing interests:** The authors declare that no competing interests exist.

## Introduction

Stem cell (SC) maintenance and function depend on signals from their local microenvironment, the SC niche. Adult neural, mesenchymal, and hematopoietic SCs neighbor the vasculature, which not only supplies oxygen and nutrients but also provides molecular signals to the stem cells (*Goldberg and Hirschi, 2009*; *Gómez-Gaviro et al., 2012*; *Oh and Nör, 2015*; *Waldau, 2015*), particularly through endothelial cells (*Azevedo et al., 2017*; *Gao et al., 2018*; *Perlin et al., 2017*). An intimate molecular communication from the vascular niche to the SCs contributes to tissue homeostasis and repair. Reverse signaling, from SCs to the neighboring niche, was also shown in zebrafish where a hematopoietic SC can indeed remodel the perivascular niche (*Tamplin et al., 2015*). Little is known about skin vasculature cross-talking with adult hair follicle stem cells (HFSCs). Previously, we proposed a hypothetical model in which adult epithelial HFSCs cluster in their tissue residence (the bulge) to produce gradients of signaling molecules that might remodel the surrounding microenvironment (*Fuchs et al., 2004*; *Tumbar et al., 2004*).

HFSCs reside in the upper segment of the hair follicle (HF) known as the bulge, and are embedded deep into the skin dermis. During hair cycle, bulge HFSCs periodically regenerate the temporary lower HF region (bulb), which grows downward into the hypodermis (*Cotsarelis, 2006*). HFs undergo morphologically recognizable and synchronous phases of remodeling, known as catagen (bulb destruction), telogen (quiescence and rest) and anagen (bulb growth). At catagen a subset of quiescent HFSCs leave the bulge, and form the secondary hair germ, which replaces the apoptotic hair bulb. The skin loses most of its fat in the hypodermis, shrinks considerably in thickness, and the HFs enter the resting phase, or telogen. In anagen, signals from the environment, including the fat progenitors (*Goldstein and Horsley, 2012*), activate the quiescent HFSCs in the hair germ to

produce a new hair bulb with a newly growing hair shaft (*Cotsarelis, 2006*). The bulge HFSCs that go into the hair germ are subsequently replenished at anagen by self-renewing symmetric divisions of the bulge SCs (*Zhang et al., 2009*; *Zhang et al., 2010*). The fat layer in the hypodermis and surrounding the growing hair bulb also regenerates in parallel with the hair follicle at anagen (*Goldstein and Horsley, 2012*).

Although poorly understood, the skin vasculature is remodeled along with the massive changes in the skin structure and thickness that result in destruction and reconstruction of the highly vascularized hypodermis. In human skin, vasculature associated cells known as pericytes may promote SC proliferation of the inter-follicular epidermis (*Zhuang et al., 2018*), although any effects on hair follicle stem cell activation have not been addressed. Furthermore, a permanent vascular structure, known as the upper venule annulus, neighbors the upper hair bulge region at all hair cycle stages (*Xiao et al., 2013*), but the significance of this association is not understood. Importantly, angiogenesis and recruitment of blood vessels promote stronger hair shaft production, presumably by providing nutrients to rapidly dividing hair matrix progenitor cells in the bulb during full anagen (*Mecklenburg et al., 2000*; *Yano et al., 2001*), but these studies have not examined the potential cross-talk of vasculature with quiescent HFSC at earlier stages of hair cycle (catagen/telogen).

Previously we showed that adult HFSC activation and telogen to anagen progression are promoted by runt-related transcription factor 1 (Runx1) (*Hoi et al., 2010*; *Osorio et al., 2008*), a transcription factor also important for hematopoietic SCs (*Yzaguirre et al., 2017*). Runx1 is up-regulated at catagen in quiescent HFSCs migrated from the bulge, which form the hair germ, where Runx1 promotes the 'primed' stem cell state (*Lee et al., 2014*; *Scheitz et al., 2012*). This cell state is more susceptible to activation from quiescence (*Gonzales and Fuchs, 2017*), because of changes in lipid metabolism and in susceptibility to specific signaling (e.g. Wnt) which we showed were driven by Runx1 (*Lee et al., 2014*; *Osorio et al., 2011*; *Scheitz et al., 2012*).

Here, we characterized in detail the arrangements of vasculature near the HFSC activation zone, and found a transient dense horizontal plexus under the hair germ during late stages of quiescence (late catagen/telogen). We show that *Cdh5-CreERT2* induced endothelial-specific mutation in *Activin A Receptor Like Type 1* (*Acvrl1*, or *Alk1*), a gene known for controlling vasculature remodeling in other systems (*González-Núñez et al., 2013*; *Oh et al., 2000*), affects the hair-cycle related remodeling of skin vasculature. This mutation results in increase of CD31+ vasculature density in the horizontal plexus near the hair germ and impaired HFSC activation with delayed progression into anagen. We provide evidence that skin endothelial cells produce BMP signals, previously proven essential for HFSC quiescence and hair cycle (*Botchkarev and Sharov, 2004*; *Lee and Tumbar, 2012*). Conversely, we find that *Runx1* mutation in the epithelium not only delays stem cell activation and hair cycle progression as we showed before, but also increases the density of vasculature in the horizontal plexus under the hair germ. Our data are consistent with a model in which increased vasculature near the HFSC activation zone is inhibitory to stem cell activation and prolongs quiescence by delaying progression from telogen into anagen. We propose that reciprocal communication and coordination between HFSCs and vasculature are essential for proper skin homeostasis and for timely HFSC activation, and outline target genes for future mechanistic studies to dissect the molecular pathways involved in this process.

## Results

### Horizontal vascular plexus under hair germ transiently neighboring hair follicle stem cell activation zone during hair cycle

To understand in detail how the skin vasculature is remodeled near the HFSC activation zone in the hair germ during hair cycle, we sacrificed C57BL/6 wild type mice at late catagen (PD19), telogen (PD20), early anagen (PD21) and anagen (PD28) (*Figure 1* and *Figure 1—figure supplement 1*). Hair cycle stages were determined by morphology and by staining for Ki67, a proliferation marker (*Figure 1—figure supplement 1*). As expected, skin thickness increased prominently from telogen to anagen due to expansion of the hypodermis and due to hair bulb growth, and the total skin area covered by CD31+ signal for vasculature also increased (*Figure 2A* and *Figure 2—source data 1*). Remarkable changes in skin vasculature organization, as marked by CD31 staining, were apparent during hair cycle in analysis of both 70 μm thick (*Figure 1*) and 10 μm thin (*Figure 1—figure*

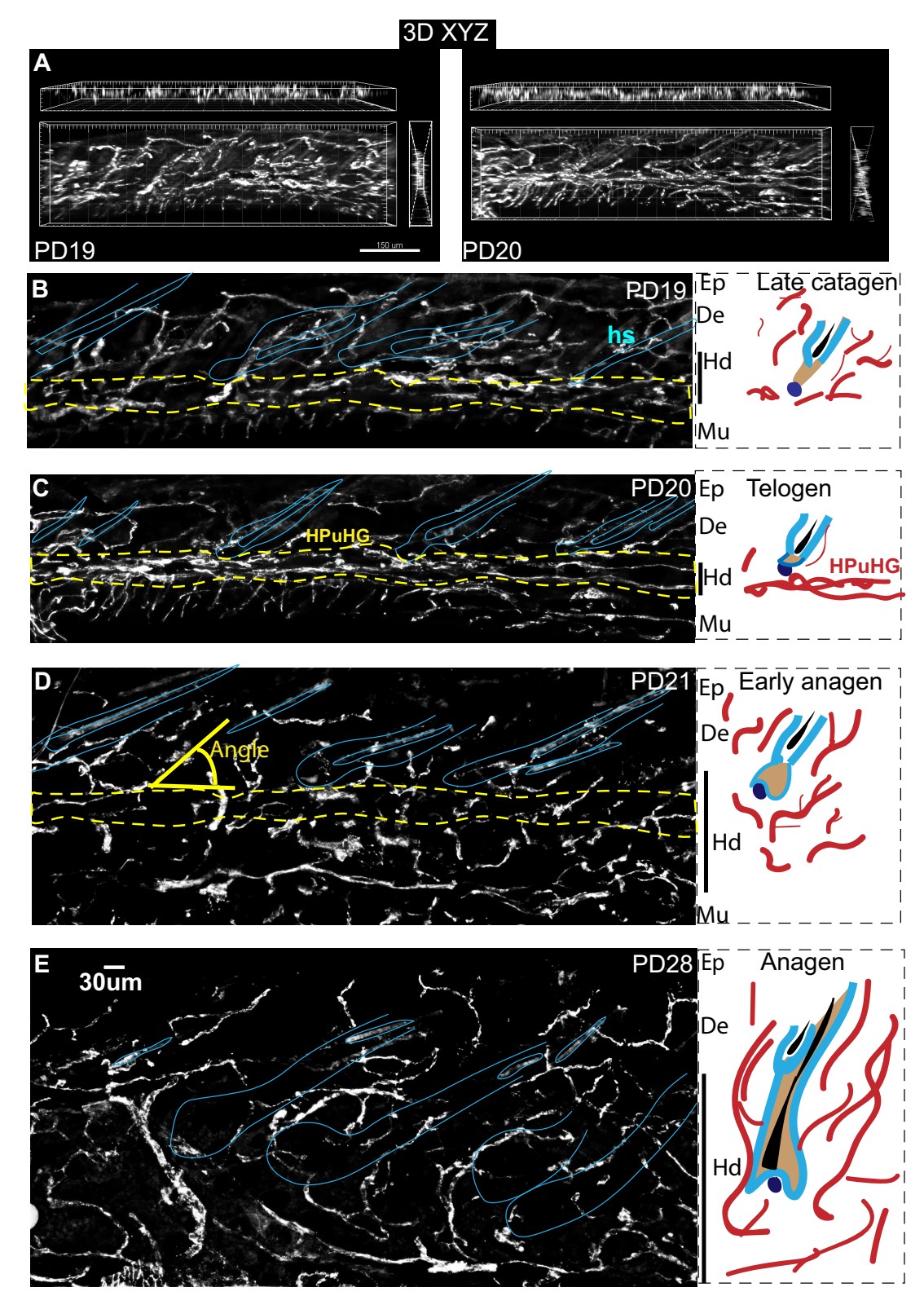

**Figure 1.** Transient horizontal plexus under hair germ (HPuHG) precedes hair follicle stem cell activation in hair cycle. (A–E). CD31 images using widefield fluorescence microscopy, with optical Z-stacks and deconvolution from 70 μm thick skin sections, shown as maximal projection images. Yellow-dotted line indicates the region of HPuHG. Solid yellow line shows the angle of vasculature branch relative to the epidermis. Corresponding region of epidermis (Ep), dermis (De), hypodermis (Hd), and muscle (Mu) are noted immediately on the right of each microscopic image. This

*Figure 1 continued on next page*

*Figure 1 continued*

demarcation is apparent in images prior to DAPI channel splitting and contrasting in Photoshop. Both the hair follicles and old hair shafts which show non-specific signal in antibody staining of skin were highlighted with light blue line. Panels on right show schematic of the hair cycle stage, which was extracted from DAPI staining of these skin sections (not shown) and from analysis of adjacent thin sections from the same mice (*Figure 1—figure supplement 1*, left panels). The hair cycle stages were confirmed by Ki67 and caspase staining of samples from adjacent skin section (*Figure 3* and *Figure 1—figure supplement 1*, right panels). In cartoons red cables represent schematic of vasculature arrangement that reveal dense horizontal plexus under the hair germ (HPuHG) at telogen. The number of mice analyzed per stage is shown in *Figure 2*. Ep, Epidermis. De, Dermis. Hd, Hypodermis. Mu, Muscle.

DOI: https://doi.org/10.7554/eLife.45977.002
The following figure supplement is available for figure 1:

**Figure supplement 1.** Imaging of thin and thick skin sections reveals remodeling of vasculature at different hair cycle stages.
DOI: https://doi.org/10.7554/eLife.45977.003

*supplement 1*) skin sections. In addition, the telogen (PD20) skin vasculature appeared more horizontal (parallel to epidermis) when compared with vasculature at late catagen (PD19) or anagen (PD21, PD28), as shown by images in *Figure 1* and *Figure 1—figure supplement 1* and by quantification in *Figure 2C*. Optical Z-sections in confocal microscopy or in wide field fluorescence with digital deconvolution and maximal projection allowed examination of 3D organization changes of skin vasculature during hair cycle (*Figure 1*). These changes are quantified from maximal projection images like those in *Figure 1B–E* and the results are summarized in *Figure 2* and described in more detail below.

Specifically, we noticed a change in the density of CD31+ vasculature underneath the hair germ during the hair cycle, with the highest measured density at PD20 (telogen) (*Figures 1* and *2B* and *Figure 2—source data 2*). We will refer to the dense horizontal vasculature structure underneath the hair germ at telogen as the horizontal (H) plexus (P) under(u) the hair (H) germ (G) (HPuHG). The data suggest that the HPuHG forms at late catagen when skin vasculature changes the orientation from vertical to horizontal as shown by our angle measurements (*Figure 2C* and *Figure 2—source data 3*), and it is most prominent by telogen (PD20). The HPuHG disperses again at early anagen (PD21) and anagen (PD28) when the vasculature regains its vertical orientation relative to epidermis, as shown by data in *Figures 1* and *2B–C*. Parallel with the morphological re-organization of skin vasculature around the hair germ described above, CD31+ endothelial cells undergo apoptosis at catagen (*Figure 3A,B*) and proliferation at anagen (*Figure 3C–F* and *Figure 3—source data 1*), as shown by immunostaining for Caspase-3 and Ki67 and quantification. Importantly, the colocalization data were analyzed in single optical confocal images to rule out confounding interpretation from possible vasculature associated cells rather than CD31+ endothelial cells. The detection of proliferation and apoptosis in skin endothelial cells during hair cycle was in line with previously reported data (*Mecklenburg et al., 2000*; *Yano et al., 2001*).

Thus, the skin endothelium and hair follicle epithelium undergo parallel cell death and proliferation during catagen and anagen respectively, accompanied by notable morphological changes in both compartments. Specifically, from late catagen to telogen when HFSCs in hair germ prepare for subsequent activation, the skin vasculature becomes more horizontal and forms a characteristic plexus, with dense vessels bundling underneath the hair germ, a structure that we call the HPuHG. The HPuHG is transient, as it is strongly present in the vicinity of the hair germ at telogen. From telogen to early anagen, when HFSCs are activated and begin to proliferate, the HPuHG is dispersed and the vasculature largely regains its vertical orientation while the hypodermis expands pushing the remaining horizontal vasculature bundles away from the vicinity of the proliferating hair germ (see *Figure 1* right panels for cartoon summary). Following this remarkable skin vasculature morphological remodeling through late catagen/telogen and early anagen, later on in full anagen the CD31+ cells also proliferate robustly undergoing angiogenesis followed by endothelial cell apoptosis at catagen (*Figure 3E,F*).

## *Alk1* endothelial-specific mutant via *Cdh5-CreERT2* increases density of HPuHG and delays HFSC activation and hair cycle progression

To understand if skin vasculature organization during telogen is relevant to subsequent HFSC activation and hair cycle progression, we aimed to perturb the characteristic remodeling from within the

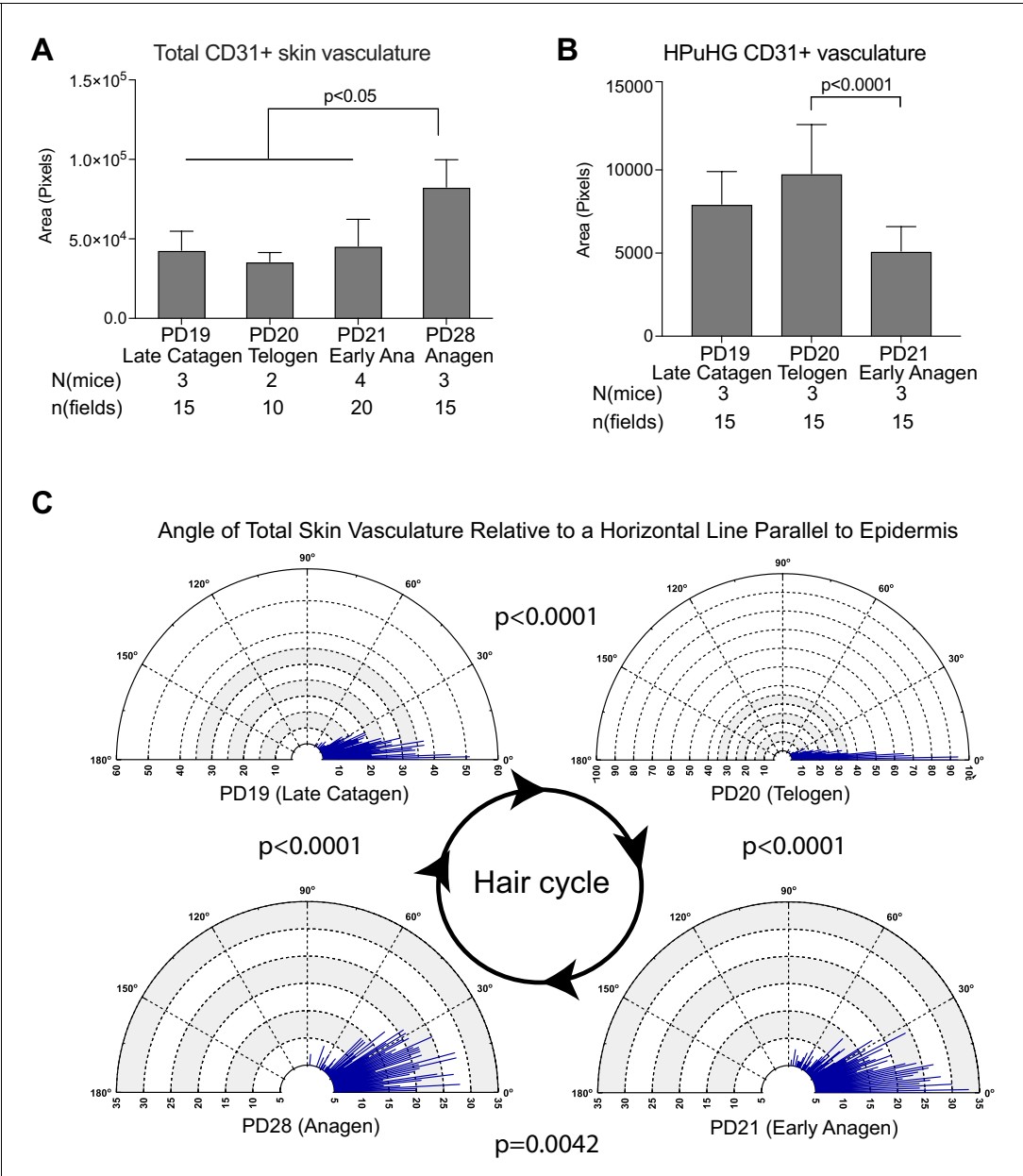

**Figure 2.** Quantification of CD31+ skin vasculature organization during hair cycle. (**A**) Quantification of total CD31+ area per skin field at different hair cycle stages (as shown in *Figure 1*) shows significant increase of vasculature at anagen. The total CD31+ area is defined by outlining CD31+ area in dermis and hypodermis using freehand selection on z-stack wide field light microscope images of 70 µm skin sections immunostained with CD31 and Dapi. in N = 5 images/mouse. Error bars represent standard deviation. (**B**) Quantification of CD31+ area in HPuHG region per skin field at different hair cycle stages shows significant decrease in the transition from telogen to anagen. HPuHG region is determined by drawing a stripe of 72-wide pixels immediately underneath the hair follicle. CD31+ area in HPuHG is determined by CD31+ area in this selected region. N = 5 images/mouse. Error bars represent standard deviation. (**C**) Distribution of angles of vasculature branches relative to the epidermis at different hair stages. Angles were measured by connecting the end point of individual vasculature branch with the nearest branch point, and measuring the relative angle of this line to a horizontal line that is parallel to the epidermis. Blue lines represent the number (count) of vasculature branches at specific angles. Count coordinates are on the radius (y-axis) of polar graph, and angular coordinates are on the circumference (x-axis). P-values from Kolmogorov-Smirnov two-sample test indicate the distribution changes from one hair cycle stage to the other. Note predominant horizontal distribution at telogen (n = 3 mice per stage, N = 5 images/mouse).

DOI: https://doi.org/10.7554/eLife.45977.004

The following source data is available for figure 2:

**Source data 1.** Spreadsheet of original quantification of CD31+ vasculature in total skin (for *Figure 2A*).

*Figure 2 continued on next page*

*Figure 2 continued*

DOI: https://doi.org/10.7554/eLife.45977.005

**Source data 2.** Spreadsheet of original quantification of CD31+ vasculature in HPuHG region (for *Figure 2B*).

DOI: https://doi.org/10.7554/eLife.45977.006

**Source data 3.** Spreadsheet of original quantification of angles of CD31+ vasculature (for *Figure 2C*).

DOI: https://doi.org/10.7554/eLife.45977.007

endothelium and examine possible effects on the hair epithelium and on hair cycle. Since nothing is known about the genetic control of vasculature remodeling we describe here during catagen/telogen/early anagen, we started by examining two endothelial-specific mutant mice, Activin A receptor like type 1 (*Alk1)* and *Neuropilin 1* (*Nrp1*), with known general functions in vasculature remodeling in other systems (*Fantin et al., 2014*; *Oh et al., 2000*). We used the previously described endothelial-specific *Cdh5-CreERT2* mouse line (*Wang et al., 2010*) crossed with either the *Nrp1^{flox/flox}* (*Nrp1* EndKO) (*Gu et al., 2003*) or *Alk1^{flox/flox}* (*Alk1* EndKO) (*Oh et al., 2000*) mice. We performed tamoxifen (TM) induction at catagen (PD17) and sacrificed mice at various time points between PD22 and PD35, when HFs are expected to be in various stages of anagen (*Figure 4A*). This specifically activated the Cre in the CD31+ endothelial cells, as shown by our confocal analysis of our *tdTomato* reporter mice (n = 3) (*Figure 4B*). We did not observe an obvious vasculature or hair cycle phenotype in the *Nrp1* endothelial-specific mutant mice by PD35 (n = 3 mice per group) (data not shown) and decided not to study it further. Although our results with *Nrp1* were negative, additional experiments are needed to definitively conclude that Nrp1 does not play a role in this context. On the other hand, *Alk1* EndKO induced with TM at PD17 (late catagen) showed an overall consistent delay in hair cycle progression into anagen at all stages analyzed in our experimental set-up (*Figure 4C–E* and *Table 1*). The delay in anagen onset was not consistent in *Alk1* End KO mice induced with TM at PD21 and PD23 after stem cell activation (*Figure 4—figure supplement 1A*, *Table 1*), suggesting a hair cycle stage specific function of *Alk1* KO, and alleviating concerns of systemic side effects of this mutation.

At all stages analyzed the *Alk1* EndKO skin from mice induced at PD17 showed a relative delay in hair cycle progression when compared to CT skin from mouse littermates. Anagen I stage is morphologically indistinguishable from telogen, and is marked by the presence of rare weakly stained Ki67+ cells in the hair germ (*Figure 4C*, bottom right panel). In PD17-induced mice sacrificed at PD25, 4 out of 5 *Alk1* EndKO mice were at telogen, whereas littermate CT mice were either at a mix of telogen and anagen I (3 out of 5 mice analyzed) or at anagen I/II (2 out of 5 mice) (*Figure 4C* left panels, and 4E). Analysis of the number of Pcad+ hair germ cells in telogen hair follicles of *Alk1* EndKO at PD25 did not show a loss of these cells relative to stage-matched CT mice in telogen at PD20 (*Figure 4—figure supplement 1B*). By PD28/31 8 of our 11 control mice were at Anagen I, II, or further along (II+), whereas 6 of our 7 *Alk1* EndKO mice were at a mix of telogen and anagen I (*Figure 4C*, right panels and *Figure 4E*). Although apparently healthy and normal in size by PD25-28, approximately two-thirds of *Alk1* EndKO mice succumbed to sudden death by PD35, likely due to hemorrhage in the lung as previously reported (*Park et al., 2009*). In the surviving PD35 KO mice, hair follicles were at anagen I/II, whereas hair follicles in control mice had progressed into anagen III/IV (*Figure 4D and E*). We conclude that loss of *Alk1* in skin endothelial cells starting at catagen (PD17) but not at early anagen (>PD21) results in subsequent impaired HFSC activation and delayed progression of hair follicles through the hair cycle stages.

To understand how *Alk1* deletion in the endothelium at late catagen affected the skin vasculature reorganization during hair cycle, we analyzed 60μm-thick skin sections stained with CD31 antibodies by confocal microscopy and optical Z-stack imaging. To avoid differences in vasculature related to hair cycle delay, we compared CT and *Alk1* EndKO mice with matched hair cycle stages at either telogen or at telogen/Anagen I, as shown by Ki67 staining (*Figure 4—figure supplement 1C*). Inspection of maximum projection stack images revealed an obvious increase in CD31+ skin vasculature of the HPuHG in *Alk1* EndKO relative to littermate CT mice at both stages analyzed (*Figure 5A*). The absolute CD31+ area encompassing the entire HPuHG structure was quantified and showed significant changes at both hair cycle stages analyzed (*Figure 5B,C* and *Figure 5—source data 1*), with abnormally high density in the *Alk1* EndKO not observed at any hair cycle stages in normal mice.

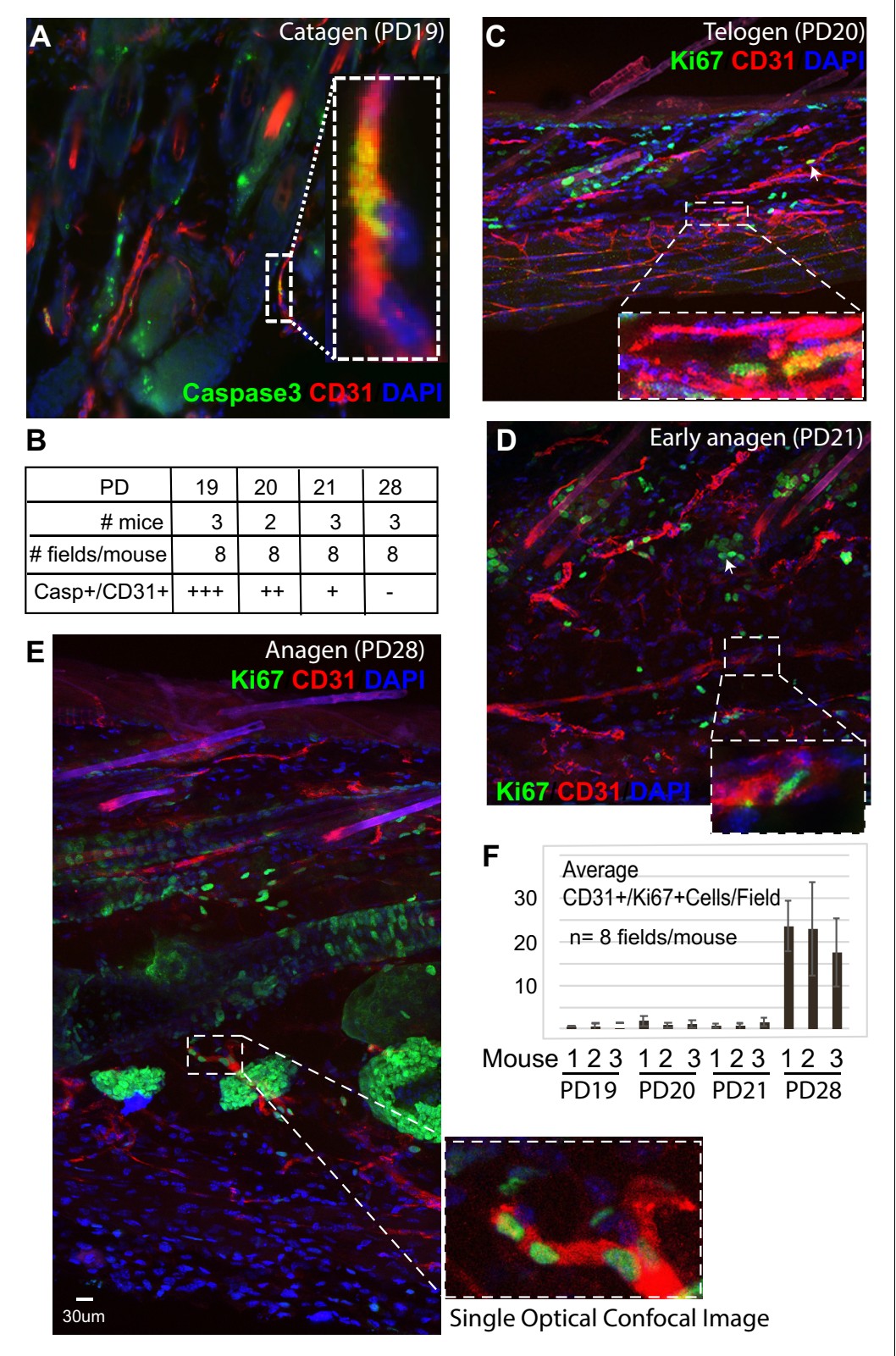

**Figure 3.** CD31+ vasculature undergoes proliferation at anagen and apoptosis at catagen. (A–F) Skin sections from mice in *Figure 1* at late catagen (PD19), telogen (PD20), early anagen (PD21), and anagen (PD28) stained for CD31 and Ki67 or activated Caspase 3, imaged using confocal microscopy and Z-stacks (A,C,D,E) and shown as maximal projections. (B, F) Quantification was done by comparing Z-stack maximal projections with individual optical slices to verify co-localization. Caspase-3 quantification (B) was difficult to count as individual cells, and is shown as a semi-quantitative measure

*Figure 3 continued on next page*

*Figure 3 continued*

where '+++" means most frequent and "– "means absent. Strongly positive Ki67 cells are peripherally associated with vasculature, but are not actually CD31+ as shown by analysis of both individual confocal optical sections and projections through stack. Double positive CD31+/Ki67+ cells display lower Ki67 signal at telogen and early anagen than the vasculature-associated cells, and are shown as enlarged and differentially enhanced insets in dotted white boxes. Note their presence at the bottom of the skin below the hair follicles. Note patches of CD31+/Ki67+ cells at full anagen (**E**) and robust proliferation of the endothelium at this stage quantified in (**F**). Error bars represent standard deviation.

DOI: https://doi.org/10.7554/eLife.45977.008

The following source data is available for figure 3:

**Source data 1.** Number of Ki67+ CD31+ double positive cells during hair cycle (for ***Figure 3F***).

DOI: https://doi.org/10.7554/eLife.45977.009

The ratio of CD31+ area over a selected fixed area underneath the hair was also measured to obtain the vasculature density under the hair germ, and it was also found increased in *Alk1* EndKO (***Figure 5—figure supplement 1A,B***); see also material and methods for more detail about quantification. The 'total skin area' under hair germ and above the muscle, that encompasses the skin hypodermis did not vary significantly in *Alk1* EndKO and CT mice, suggesting that the thickness of the fat layer was not affected (***Figure 5—figure supplement 1C,D***).

To test how arteries, which carry oxygen and nutrients to the tissue, may be affected by Alk1 EndKO, we stained PD28 skin for Ephrinb2, a marker of arteries (***Rocha and Adams, 2009***) (***Figure 5D***) and used neurofilament 1 (NF1) as a marker to exclude nerves that may also be Ephrinb2 positive. Strikingly, we observed that the HPuHG is filled with CD31$^{high}$Ephrinb2$^{low}$ vessels, thus non-arteries. This finding is significant because it suggested that *Alk1* deletion-induced changes in HPuHG vasculature may not primarily perturb the supply of oxygen and nutrients to the follicle. The CD31$^{low}$Ephrinb2$^{high}$NF1$^{negative}$ vessels (e.g arteries), appeared largely absent from the HPuHG, were clearly present in the CT and the *Alk1* EndKO skin in the subcutaneous muscle region below the HPuHG indicating our staining worked (***Figure 5D***). Though analysis of these mutant mice at later stages awaits further analysis, our brief inspection of more advanced PD31 skin found at anagen I/II suggested that in the *Alk1* EndKO the non-arterial vascular structures were visibly increased throughout the entire skin, and were now branched vertically similar with control skin (***Figure 5—figure supplement 1E***). Pending further analysis, this result suggested that the remodeling of vasculature progressed, albeit slowly, in the *Alk1* EndKO.

We conclude that *Alk1* is important in the skin endothelium for proper reorganization of vasculature during late catagen and telogen stages of the hair cycle. The effect on the epithelium was hair cycle specific, and was not observed in mice induced with tamoxifen at PD21-22. In the absence of *Alk1* in endothelium during the catagen/telogen transition, we found prominent increase in the density and the persistence of the horizontal plexus under hair germ (HPuHG). This was accompanied by impaired HFSC activation in the hair germ and by general delay in the hair cycle stage progression.

## Skin vasculature produces BMP4, a signal known essential for HFSC quiescence and hair cycle

The data so far suggested a correlation between high density of vasculature (non-arteries) underneath the hair germ at late catagen/telogen with prolonged HFSC quiescence and delay in hair cycle stage progression. This differs from previous work in which increasing anagen vasculature by promoting angiogenesis would simply induce more rapid hair shaft growth through increased oxygen and nutrients (***Yano et al., 2001***). Our results instead suggest that prior to HFSCs activation and anagen onset, at late catagen/telogen, increased non-arterial vasculature near the hair germ has an inhibitory effect on stem cell activation and hair growth. Therefore, we considered the possibility that endothelial cells in the HPuHG may secrete HFSC quiescence-inducing factors, of which the most well-understood candidate is BMP (***Botchkarev and Sharov, 2004***; ***Lee and Tumbar, 2012***). The choice of this candidate was also justified by previous studies in lung, which demonstrate that endothelial cells express BMP4 (***Frank et al., 2005***). BMP4 protein and mRNA levels oscillate through the hair cycle and are highly produced in skin during quiescence, whereas *BMP Receptor Type-I (BMPRI)* epithelial deletion is sufficient to activate the HFSCs (***Botchkarev and Sharov, 2004***; ***Lee and Tumbar, 2012***).

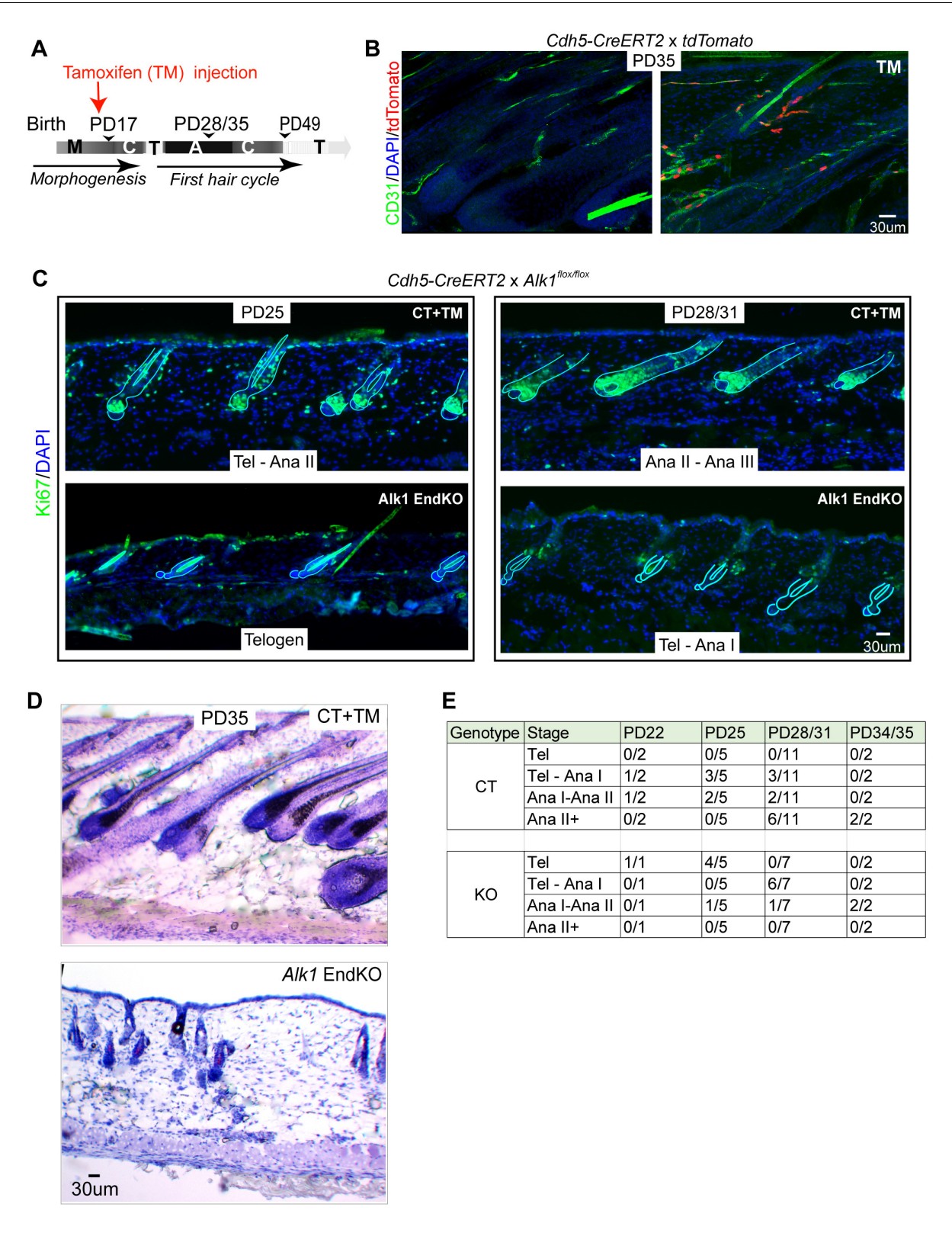

**Figure 4.** Endothelial-specific *Alk1* knockout delays HFSC activation and hair cycle. (**A**) Schematic of tamoxifen (TM) inductions using the endothelial-specific *Cdh5-CreERT2*, with hair cycle stages indicated at postnatal days. M, morphogenesis; C, catagen, T, telogen, A, anagen. (**B**) Skin sections from *Rosa26-tdTomato* reporter mice crossed with *Cdh5-CreERT2*, induced with tamoxifen (TM) at PD17, and sacrificed at PD35, show specific induction in the CD31+ vasculature. (**C**) Skin sections from *Alk1*<sup>flox/flox</sup> mice crossed with *Cdh5-CreERT2* and induced with tamoxifen (TM) at PD17 show different

*Figure 4 continued on next page*

*Figure 4 continued*

hair cycle stages as indicated by morphology and staining for Ki67 at PD25 and at PD28/31. (**D**) Same as (**C**) but stained with hematoxylin and Eosin and mice were sacrificed at PD35. (**E**) Number of littermate control (CT) *Alk1* End KO (KO) mice at specific indicated hair stage obtained from Ki67 staining/ total number of mice analyzed at each age/genotype group. For instance, at PD25 we analyzed a total of 5 control (CT) mice of which three were at Tel-Anal and two were at Ana I-II. Also see *Table 1* for more detailed information on each mouse and its respective hair cycle stage.

DOI: https://doi.org/10.7554/eLife.45977.010

The following figure supplement is available for figure 4:

**Figure supplement 1.** Phenotype of *Alk1* endothelial knockout skin.

DOI: https://doi.org/10.7554/eLife.45977.011

To examine if BMP4 protein is indeed expressed in skin vasculature we stained skin sections for BMP4 and CD31 from both WT and *Alk1* EndKO mice that were hair cycle stage matched, at Telogen/Anagen I, when rare Ki67+ cells are present in some HFs. Confocal imaging revealed pronounced BMP4 signal in vasculature-like structures, which expressed variable levels of CD31. These structures were distributed throughout the skin, but their density was increased in the HPuHG region (*Figure 6A*, left panels). The BMP4 expression was further accentuated in the HPuHG region in the *Alk1* EndKO mice (*Figure 6A*, right panels and quantified in 6B and *Figure 6—source data 1*). Some BMP4+ vasculature-like structures of the HPuHG show lower or possibly absent levels of CD31, and these were more pronounced in the *Alk1* EndKO. The basis of this phenotype will be examined more in future, but we suspect it has to do with a reduction of CD31 expression in the *Alk1* EndKO vasculature. Quantification of BMP4 signal in the CD31+ regions in HPuHG area confirmed our visual assessment that CD31+ endothelial cells do not produce more BMP4 in the *Alk1* EndKO skin (*Figure 6C* and *Figure 6—source data 1*). Taken these considerations together, the increase in BMP4 signal in the HPuHG area can be reasonably attributed to increased density of vasculature in this region. Further analysis showed that the interfollicular epidermis (IFE) expressed equal BMP4 levels in both control and *Alk1* EndKO skin (*Figure 6D* and *Figure 6—figure supplement 1A* and *Figure 6—source data 1*). The increase in BMP4 levels in skin from the *Alk1* EndKO mice relative to controls was also supported by our total skin Western blots and quantification (*Figure 6E,F* and *Figure 6—figure supplement 1B,C*). We conclude that skin vasculature produces BMP4, and thus its clustering in the HPuGH at late catagen/telogen generates high concentration of HFSC-quiescence inducing factors underneath the HFSC activation zone. These correlations provide a plausible, though likely partial, explanation of the delay in HFSC activation and hair cycle progression in the *Alk1* EndKO mice.

## Varying Runx1 levels in the epithelium affect hair cycle-related vasculature remodeling near hair follicle stem cell activation zone

The data presented so far suggested a possible communication from vasculature to promote the quiescence of HFSCs prior to activation and anagen onset. Parallel experiments in our laboratory suggested that the opposite communication from epithelium towards the endothelium may also be true, whereby gene expression in quiescent epithelial HFSCs may influence vasculature remodeling during hair cycle. This idea emerged from our studies of the transcription factor Runx1, which we previously found that when deleted it delays HFSC activation and anagen onset (*Hoi et al., 2010*; *Osorio et al., 2008*). Intriguingly, we had previously found by genomic studies that Runx1 expression in the HFSCs resulted in mRNA level changes in genes encoding secreted molecules with known roles in vasculature remodeling, such as: *Sema3a*, *Figf*, *Ntn4*, *Adamts1*, *Cxcl1*, *Cxcl10*, *Cxcl12*, *Cx3cl1*, *Edn1*, *Col5a1*, and *Pla2g7* (*Table 2*) (*Lee et al., 2014*). These changes in mRNAs occurred by doxycycline-induced *Runx1* expression in quiescent bulge HFSCs within one day of administration, attesting to rapid induction of downstream target gene expression (*Lee et al., 2014*). This intriguing observation suggested the possibility that HFSCs use Runx1 with its target genes encoding secreted proteins from the epithelium to remodel vasculature around the hair follicle.

To examine the possibility that epithelial Runx1 may drive remodeling of vasculature around the hair follicle during hair cycle, we examined CD31 signal distribution around mutant hair follicles with varying Runx1 levels during late stages of hair cycle quiescence (*Figure 7A,B*). It is important to emphasize that the stages analyzed here precede the *Runx1* deletion-induced delay in anagen onset, thus do not reflect variations in vasculature organization due to differences in hair cycle stages. We

**Table 1.** *Cdh5CreERT2 x Alk1^{flox/flox}* KO (EndKO) mice and hair cycle

| Mouse ID | Genotype | Sex | Hair cycle stage |
|---|---|---|---|
| **PD17-20** | | | |
| F520.01 | CT | M | Tel |
| F520.04 | CT | F | Tel |
| F520.05 | CT | M | Tel |
| F520.06 | CT | F | Tel |
| **PD17-22** | | | |
| F167.02 | CT | M | Ana I |
| F64.02 | CT | F | Tel - AnaI |
| F64.01 | KO | F | Tel |
| **PD17-25** | | | |
| F81.06 | CT | F | Tel- Ana I |
| F146.05 | CT | M | Ana I - IIa |
| F167.05 | CT | M | Ana I - IIa |
| F433.03 | CT | F | Tel- Ana I |
| F433.04 | CT | F | Tel- Ana I |
| F81.07 | KO | F | Tel |
| F86.02 | KO | F | Ana I |
| F86.04 | KO | M | Tel |
| F99.01 | KO | M | Tel |
| F433.06 | KO | F | Tel |
| **PD17-28** | | | |
| E963.02 | CT | M | Ana IIIc |
| F80.04 | CT | F | Tel - Ana I |
| F90.01 | CT | M | Ana I - Ana IIa |
| F146.02 | CT | F | Ana IIa-c |
| F333.01 | CT | M | Ana I - IIa |
| F333.03 | CT | M | Tel - Ana I |
| F333.04 | CT | M | Tel - Ana I |
| F11.04 | KO | F | Tel - Ana IIa |
| F89.01 | KO | M | Ana I-IIa |
| F80.03 | KO | M | Tel - Ana I |
| F90.02 | KO | F | Tel - Ana I |
| F333.02 | KO | F | Tel - Ana I |
| **PD17-31** | | | |
| F3.01 | CT | M | Ana II - Ana III |
| F3.03 | CT | M | Ana II |
| F90.05 | CT | M | Ana II-III |
| F146.06 | CT | M | Ana III |
| E906.01 | KO | M | Tel-Ana I |
| F90.07 | KO | F | Tel-Ana I |
| **PD17-34/35** | | | |
| E928.03 | CT | F | Ana III |
| E928.05 | CT | F | Ana IV |
| E960.01 | KO | F | Ana I - Ana II |

*Table 1 continued on next page*

*Table 1 continued*

| Mouse ID | Genotype | Sex | Hair cycle stage |
|---|---|---|---|
| E928.01 | KO | F | Ana I-Ana II |
| **PD21-26** | | | |
| F218.01 | CT | M | Ana I - Ana III |
| F218.05 | KO | M | Ana III |
| **PD21-31** | | | |
| F218.02 | CT | F | Ana I - Ana II |
| F218.04 | KO | F | Tel - Ana I |
| **PD23-26** | | | |
| F273.02 | CT | F | Ana III |
| F273.01 | KO | M | Ana III |

DOI: https://doi.org/10.7554/eLife.45977.012

employed previously generated mutant mice with epithelial deletion of *Runx1 (Krt14-CreERT2* x *Runx1flox/flox*, which we refer to as *Runx1* EpiKO) in which we have demonstrated that 97% of HFs lose Runx1 expression by PD20 when tamoxifen is injected at PD17 (*Scheitz et al., 2012*). Runx1 protein is depleted from hair follicle within 24 hr post tamoxifen induction demonstrating its rapid turnover (*Hoi et al., 2010*), a phenomenon we also observed with another transcription factor, Gata6 (*Wang et al., 2017*). Conversely, we previously demonstrated that *Runx1* is rapidly induced in the epithelium of our *Krt14-rTA* x *pTRE-Runx1* (*Runx1* EpiTG) and induces mRNA changes in vasculature-related genes within 24 hr of doxycycline induction (*Lee et al., 2014* and *Table 2*). Here *Runx1* EpiKO and EpiTG mice were induced with tamoxifen (TM) or doxycycline (doxy) respectively at postnatal day (PD) 17, when HFs are at late catagen and vasculature undergoes dramatic remodeling. By PD20 we expected the hair follicles would be at catagen VIII/telogen and the vasculature would have undergone its remodeling at late catagen.

Skin sections from *Runx1* epithelial mutant mice and littermate controls were stained for vasculature marker CD31 (*Figure 7C*). HFs were carefully matched for hair cycle stage, and in our first approach to examine vasculature around the hair germ we simply measured the length and frequency of vasculature neighboring the HF bulge and hair germ. This analysis revealed significant and contrasting changes in the extent to which vasculature is found around HFs in the two *Runx1* epithelial mutants, when compared with their respective littermate controls (*Figure 7D–E*). Visual inspection of skin sections and images further suggested that the vasculature below the hair germ was denser, more aggregated in the *Runx1* EpiKO mouse (*Figure 7C*). This prompted us to inquire into the late catagen/telogen reorganization of the HPuHG in the *Runx1* EpiKO skin using thick (60 µm) skin sections with optical Z-section imaging and reconstruction of larger skin areas encompassing multiple hair follicles, as we did for the *Alk1* EndKO. Significantly, inspection of maximal projection images showed noticeable denser HPuHG in the *Runx1* EpiKO skin relative to controls (*Figure 8A,B* and *Figure 8—figure supplement 1A*). To quantify this, we defined a 'selected region' of interest within a fixed distance from hair germ and encompassing the HPuHG, and observed an increase in CD31 vasculature signal in this region in the *Runx1* EpiKO when compared with littermate controls. Quantification showed that this increase was true both at catagen VIII (*Figure 8C* and *Figure 8—source data 1*) and at telogen (*Figure 8—figure supplement 1B*). The HPuHG density below the hair germ, expressed as ratio of CD31+ vasculature divided by the area of 'selected region' was also significantly increased by epithelial *Runx1* deletion (*Figure 8—figure supplement 1C*). Whereas the thickness of the hypodermis (total area under HG) was variable from image to image even within the same mouse, we found that overall the size of hypodermis did not significantly differ between *Runx1* EpiKO and control mouse groups (*Figure 8—figure supplement 1D*). Furthermore, Restricted Maximum Likelihood (REML) statistical analysis showed that the differences in HPuHG between control and *Runx1* EpiKO were independent of the 'total skin area' (size of hypodermis) under the hair germ (*Figure 8D* and *Figure 8—figure supplement 1D,E* and *Figure 8—source data 1*). These data ruled

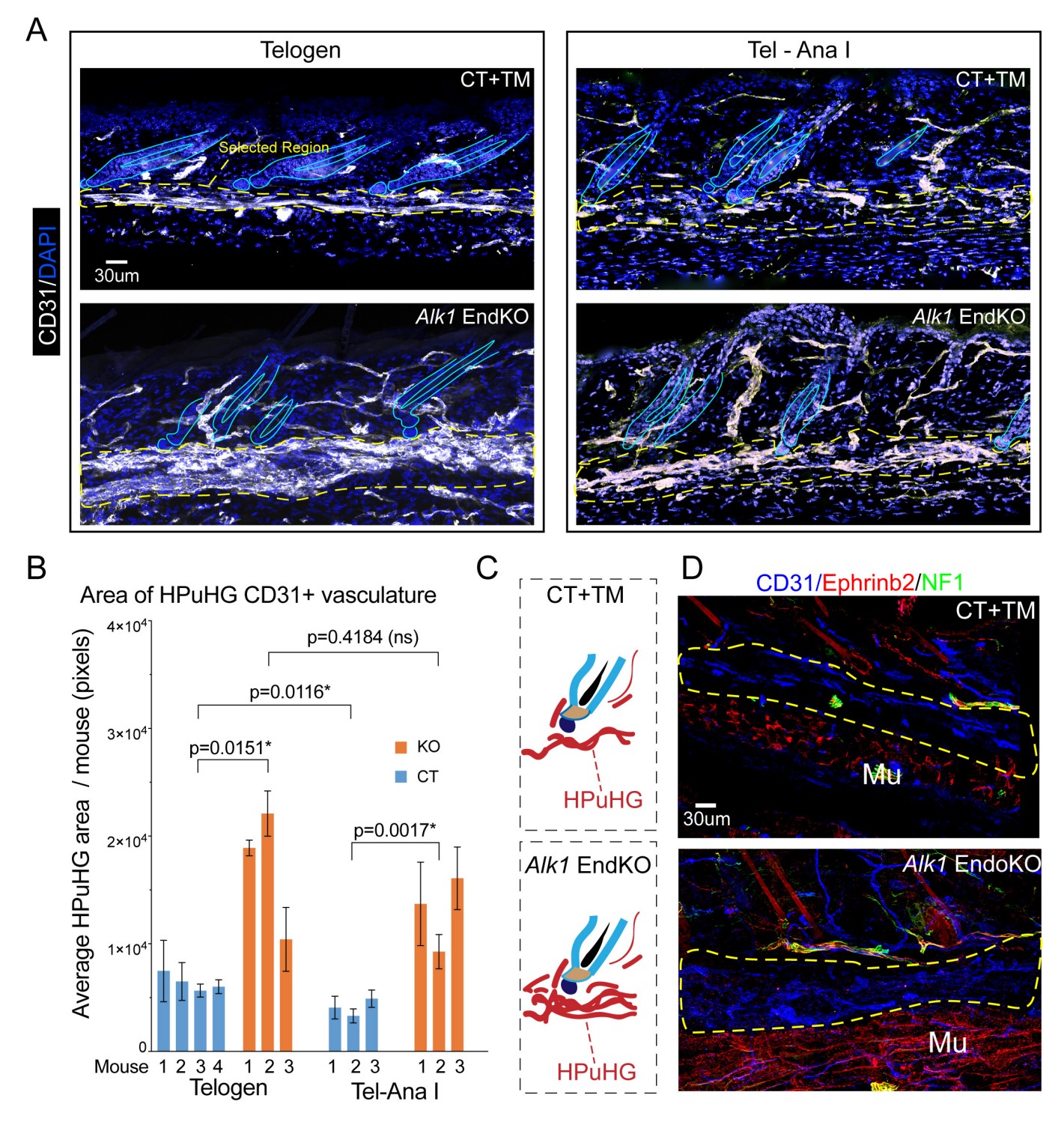

**Figure 5.** Increased vasculature density in *Alk1* endothelial knockout skin in horizontal plexus under the hair germ (HPuHG). (**A**) Skin sections from *Alk1^flox/flox^* mice with *Cdh5-CreERT2* (EndKO) or without (CT) were injected with tamoxifen (TM) at PD17 and sacrificed at different ages between ~PD 22-PD28, were stage matched in two groups: Telogen (no KI67+ cells in the hair germ and Tel-AnaI (when some hair follicles show rare Ki67 cells in the hair germ). Images are maximum projections through confocal microscope optical Z-stacks of 60 μm sections. Note visibly denser cables of CD31+ vasculature in the horizontal plexus under the hair germ (HPuHG) in the *Alk1* EndKO. (**B**) Quantification in 'selected region' under the hair germ (shown in yellow dotted line in A), which encompass the characteristic HPuHG structure, revealed an increase in CD31+ vasculature area in HPuHG in the *Alk1* EndKO skin. CD31+ vasculature in the HPuHG was manually defined in each image and the area in each image was plotted as average per mouse with

*Figure 5 continued on next page*

*Figure 5 continued*

error bars among individual images. REML approach was used to obtain p values between mice of the two genotypes. See also *Figure 5—figure supplement 1A–D* for more details and additional quantifications. Also note a decrease in HPuHG area between Telogen and Tel-Ana I in CT mice (p=0.0116), as also noted in our analysis of WT C57BL/6 mice in *Figure 1 and 2*. In contrast to CT mice, *Alk1* EndKO mice maintain an abnormally dense HPuHG that does not change significantly between stages analyzed (p=0.4184). N = 4–6 images per mouse. Error bars represent standard deviation. (C) Schematic of vasculature arrangement in CT+TM and *Alk1* EndKO. (D) Skin section of stage-matched *Alk1* CT+TM and KO mice at Telogen – Anagen I transition stage (when rare Ki67+ cells are present in hair germ). Stainings for CD31+ (general vasculature), Ephrib2 (arteries) and NF1 (nerves) show prominent non-arterial vasculature (Eprinb2$^{low}$CD31$^{high}$) in the HPuHG that is increased in the *Alk1* EndKO. Note strong arterial presence (Eprinb2$^{high}$CD31$^{low}$) in the region below the HPuHG, which contains the subcutaneous muscle.
DOI: https://doi.org/10.7554/eLife.45977.013

The following source data and figure supplement are available for figure 5:

**Source data 1.** Spreadsheet of original quantification of HPuHG CD31+ vasculature in Alk1 EndKO and CT (for *Figure 5B*).
DOI: https://doi.org/10.7554/eLife.45977.015
**Figure supplement 1.** Increase in CD31+ vasculature density in horizontal plexus under the hair germ (HPuHG) in *Alk1* endothelial knockout.
DOI: https://doi.org/10.7554/eLife.45977.014

out the possibility that *Runx1*-induced differences in vasculature in the HPuHG are secondary effects related to hypodermis size.

As we observed increased HPuHG vasculature in *Runx1* EpiKO skin, we also stained for BMP4 and CD31 to see if there was an increase in BMP4 near the HFSC activation zone in the hair germ. Similar to *Alk1* EndKO, BMP4 signal co-localized with CD31 signal, and BMP4 level was elevated in the HPuHG region (*Figure 8E* and *Figure 8—source data 1*). Quantification of BMP4 intensity in CD31+ vasculature did not show statistical differences between *Runx1* EpiKO and CT (*Figure 8F* and *Figure 8—source data 1*) consistent with the interpretation that increased vasculature density near the hair germ accounts for elevated BMP4 in this region.

Finally, we tested whether the effect of epithelial *Runx1* mutation on the vasculature persists in the *Runx1*-deletion mice at later stages of hair cycle past telogen. Thus, we collected skin from *Runx1*-deletion mice (*Krt14-Cre* x *Runx1$^{flox/flox}$*) (*Osorio et al., 2008*) once they eventually entered anagen, and compared them with skin sections from controls that were stage matched. However, we found no notable differences in the CD31+ vasculature at anagen (*Figure 8—figure supplement 1F*). We conclude that during catagen VIII/telogen, Runx1 acts in the epithelial HFSCs destined to be activated (hair germ) to guide the remodeling of vasculature at this stage and promote a more dispersed vasculature plexus underneath the hair germ (HPuHG). Importantly, this paracrine function of epithelial Runx1 in vasculature organization is accompanied by a delay of HFSCs activation and progression of hair cycle into anagen (*Hoi et al., 2010*; *Osorio et al., 2008*).

## Discussion

### Vasculature and hair follicle stem cell cross-communication during quiescence for proper subsequent skin homeostasis

Cross-talking between tissue stem cells and their environment is critical for tissue homeostasis. Hair follicle is an excellent system to study adult tissue regeneration, yet surprisingly little is known about the cross-communication between vasculature and HFSCs. Here we provide the first evidence for reciprocal communication that exists between vasculature and HFSCs in the hair germ during late quiescence stages of hair cycle, a stage, skin location, and specific question not examined by previous work of vasculature in the skin (*Mecklenburg et al., 2000*; *Xiao et al., 2013*; *Yano et al., 2001*; *Zhuang et al., 2018*). We demonstrate that targeted mutations to either the epithelial (*Runx1*) or the endothelial (*Alk1*) compartments during the critical stage of late catagen/telogen has reciprocal effects on proper reorganization and functionality of the opposite compartment. In both cases, the mutations delayed the subsequent timing of HFSC activation and hair cycle progression and resulted in increased density of vasculature underneath the hair germ at telogen. We describe a precise choreography of vasculature remodeling around the HFSCs activation zone (hair germ) during late quiescence phases of late catagen/telogen, with transient dense vasculature under the hair germ, that has not been previously described.

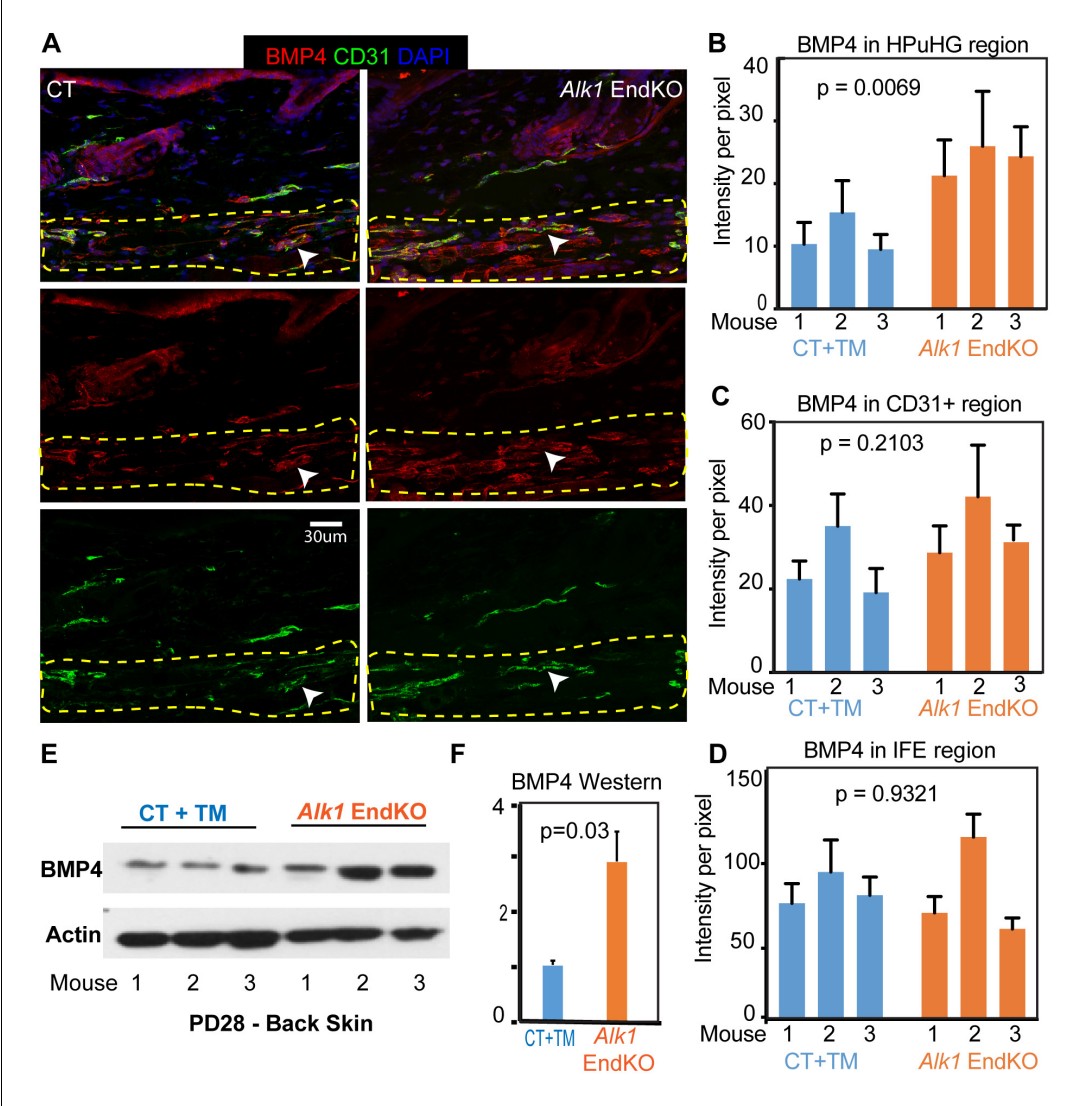

**Figure 6.** Increased BMP4 protein in the endothelial *Alk1* KO skin underneath the hair germ. (A) Representative images of skin stained with BMP4 antibodies show prominent expression in the HPuHG region in vessel-like structure. Note increased BMP4 in the HPuHG region in the *Alk1* EndKO skin. BMP4 vessel-like structures with low CD31 staining in the HPuHG may suggest possible molecular heterogeneity in vasculature in this region. (B) Quantification of average BMP4 intensity in selected HPuHG region. HPuHG region is determined by drawing a stripe of 72 pixels immediately underneath the hair follicle. Its intensity was measured in ImageJ and then subtracting average background intensity from three randomly selected background areas to get the final intensity value. *Alk1* EndKO have significantly higher intensity compared to CT+TM. P-value was obtained using REML approach in the JMP software. See also *Figure 6—figure supplement 1A* for additional images. N = 9–16 images/mouse. Error bars represent standard deviation. (C) Quantification of BMP4 intensity in CD31+ vasculature in HPuHG region indicates no significant difference between CT+TM and *Alk1* EndKO. CD31+ area was outlined by freehand tool in ImageJ and intensity was measured. The final intensity was calculated by subtracting average background intensity. P-value was obtained using REML approach in the JMP software. N = 9–16 images/mouse. Error bars represent standard deviation. (D) Quantification of BMP4 intensity in IFE region (see *Figure 6—figure supplement 1A*) indicates no significant difference between CT+TM and *Alk1* EndKO. P-value was obtained using REML approach. N = 8–10 images/mouse. Error bars represent standard deviation. (E) Western blot of back skin tissue showed increased BMP4 expression in the *Alk1* EndKO. See also *Figure 6—figure supplement 1B,C* for additional data. (F) Quantification of western blot in (E) showing BMP4 signal normalized to actin signal in each sample.
DOI: https://doi.org/10.7554/eLife.45977.016

The following source data and figure supplement are available for figure 6:

**Source data 1.** Spreadsheet of original quantification of BMP4 in Alk1 EndKO skin (for *Figures 6B, C and D*).
DOI: https://doi.org/10.7554/eLife.45977.018
**Figure supplement 1.** BMP4 levels in *Alk1* EndKO skin.
DOI: https://doi.org/10.7554/eLife.45977.017

**Table 2.** Epithelial-specific Runx1-target genes in hair follicle stem cells with known function in vasculature remodeling.

| Gene abbreviation | Full gene name | Known function | Fold change bu Runx1 TG/WT | Fold change in WT HG vs bu |
|---|---|---|---|---|
| Figf | c-fos induced growth factor | Agiogenesis, VEGF signaling | 2x | 16x |
| Col5a1 | Colagen five type a1 | Ehlers-Danlos syndrome with lethal arterial events | −11x | −2.7x |
| Adamts1 | A disintegrin-like and metallopeptidase thrombospondin type 1 | Negative regulation of angiogenesis | 2x | 2x |
| Cxcl1 | chemokine (C-X-C motif) ligand1 | Inflammatory response | 2x | 3x |
| Cxcl10 | chemokine (C-X-C motif) ligand10 | Negative regulation of angiogenesis | 6x | 16x |
| Edn1 | endothelin 1 | Patterning of blood vessels | 2x | 2x |
| Cxcl12 | chemokine (C-X-C motif) ligand 12 | Patterning of blood vessels | 2x | 2x |
| Cx3cl1 | chemokine (C-X3-C motif) ligand1 | Angiogenesis | 2x | 4x |
| Cxcl9 | chemokine (C-X-C motif) ligand9 | Inflammatory response | 7x | 10x |
| Pla2g7 | phospholipase A2, group VII (platelet-activating factor acetylhydrolase, plasma) | Inflammatory response | 10x | 6x |
| Sema3a | semaphorin 3A | Blood vessel patterning | 9x | 3x |

DOI: https://doi.org/10.7554/eLife.45977.019

Our data suggest a hypothetical model of how the HF epithelium and the skin endothelium communicate with each other (*Figure 9*), for future experimental testing. Specifically, during late catagen the vasculature changes its vertical orientation and even distribution in the skin, and transiently forms a dense horizontal plexus underneath the hair germ (HPuHG), closely neighboring the HFSC activation zone during telogen. Suggestively, we find that CD31+ skin vasculature produces BMP4 which is highly present in the HPuHG, thus generating a high concentration of this well-known HFSC-quiescence inducing factor (*Botchkarev and Sharov, 2004*; *Lee and Tumbar, 2012*) underneath the stem cell activation zone in the hair germ. BMP ligand expression levels are known to oscillate in skin through the hair cycle, inhibiting HFSC activation and hair cycle progression from telogen to anagen (*Botchkarev and Sharov, 2004*; *Lee and Tumbar, 2012*). BMP ligand expression has been documented in several different skin cell types such as fibroblast and adipocytes and reviewed in *Botchkarev and Sharov (2004)* and *Lee and Tumbar (2012)*, although BMP4 expression in skin vasculature has not been previously reported to our knowledge. *BMPRI* deletion in the epithelium is known sufficient to induce HFSC proliferation and anagen onset (*Botchkarev and Sharov, 2004*; *Lee and Tumbar, 2012*). Thus, our data showing increased density of vasculature in the HFSCs activation zone, together with previous work on BMP signaling in skin, are compelling in suggesting that the HPuHG may inhibit HFSC activation through secretion of HFSC-quiescence inducing molecules, such as BMP. Additional quiescence-inducing mechanisms of the vasculature, such as hypoxia, are seen for instance in the hematopoietic systems (*Boulais and Frenette, 2015*) and should be analyzed here. Hypoxia is especially a possible candidate, given the high density of non-arterial (non O2 carrying) vasculature in the HPuHG (this work). Our data using *Runx1* epithelial and *Alk1* endothelial-specific mutations demonstrate reciprocal communication and perturbed remodeling in the opposite compartments at late catagen/telogen. These results provide a consistent and compelling correlation between increased density of the HPuHG with prolonged HFSC quiescence that corroborate with our working model (*Figure 9*). We propose that the density of the HPuHG controls timing and robustness of subsequent HFSCs activation through BMP and other inhibitory signals, whereas the epithelium clearly modulates the density of the HPuHG forming at late stages of quiescence in hair cycle likely through specific Runx1 downstream targets relevant to vasculature (*Figure 9*).

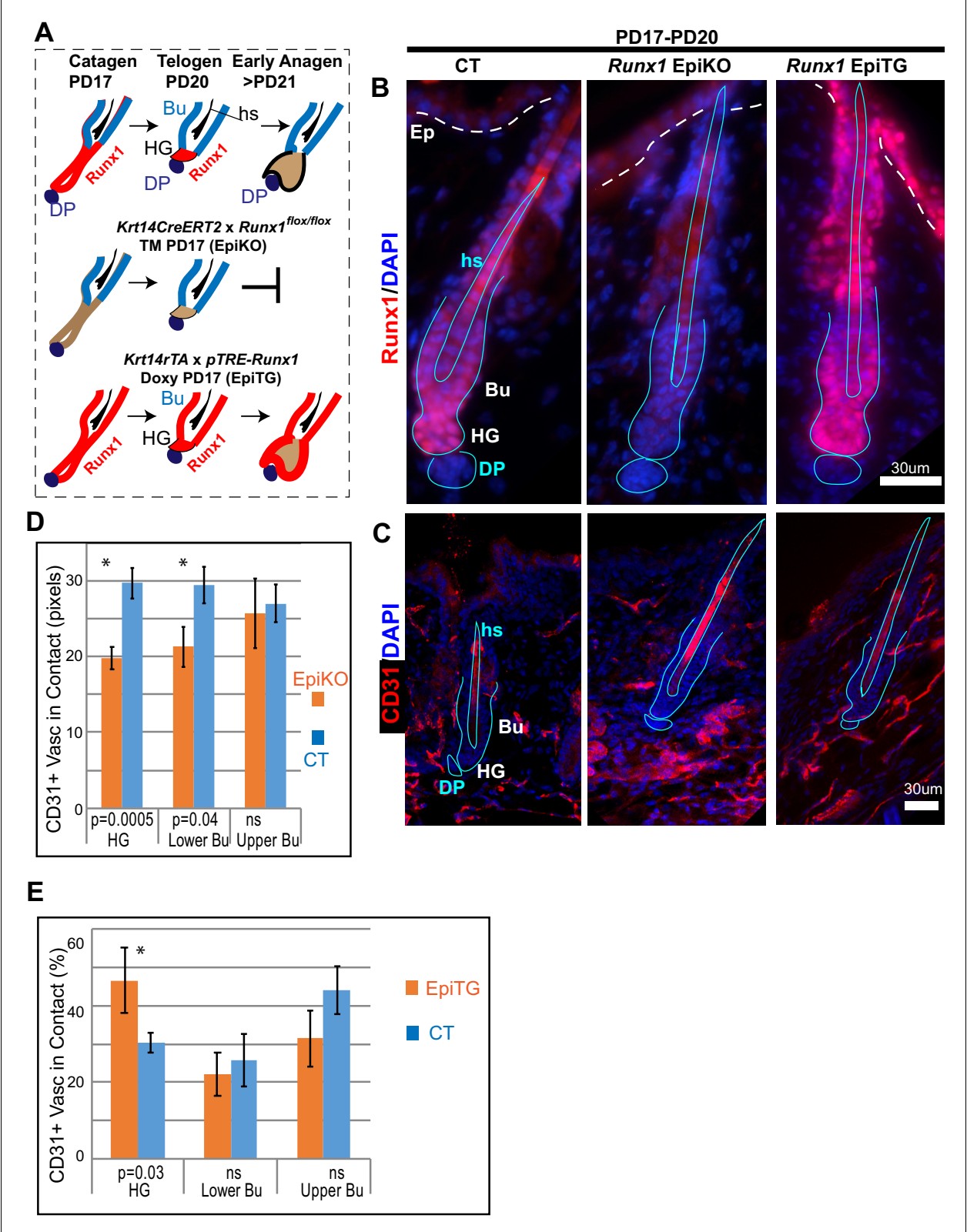

**Figure 7.** Varying level of the transcription factor Runx1 in skin epithelium results in rearrangement of CD31+ vasculature around the hair follicle. (**A**) Schematics of *Runx1* mutant inductions in genetic mouse models to create varying levels of Runx1 in the hair follicle (HF). *Runx1* expression in HF is shown in red in wild type follicles (top) and in mutant mice. Bulge (Bu) is residence of quiescent hair follicle stem cells (HFSCs). Hair germ (HG) forms at late catagen/telogen from migrating bulge HFSC and under Runx1-influence adopt the 'primed' HFSCs state. The hair germ cells are activated to

*Figure 7 continued on next page*

*Figure 7 continued*

proliferate by signals from environment and execute a new hair growth cycle in anagen. DP, dermal papillae. Hs, hair shaft. (B) Skin sections immunofluorescence-stained for Runx1 show noticeable signal in HG in normal conditions, lack of signal in EpiKO mice after TM induction, and broader expression patterns including the upper hair follicle and epidermis (Ep) in EpiTG mice, 4 days after TM and doxy induction. We have previously documented in detail and reported the high efficiency of our induction experiments (*Hoi et al., 2010*; *Lee et al., 2014*; *Scheitz et al., 2012*). (C) Staining for CD31 reveals patterns of vasculature arrangement in control and mutant mice (genotype shown at top of panel B). Note aggregated vasculature below hair germ lacking direct contact with the hair germ in EpiKO and more dispersed vasculature contacting the hair germ in the EpiTG. (D–E) Quantification of data in C reveals differences in the organization of vasculature around the hair germ, as defined by length of contact (in pixels) and frequency of contact between CD31+ endothelial cells and the epithelium. N = 4 mice per group with 40–60 hair follicle in each genotype matched for hair cycle stage at telogen. Student T test of the WT and mutant groups showed significant differences in average contact length (D) and in frequency of CD31+ vasculature around the hair follicle found in contact with the hair germ (E). Controls were distinct groups of littermate mice from either EpiKO or EpiTG mouse crossings.

DOI: https://doi.org/10.7554/eLife.45977.020

## Alk1 expression in endothelial cells is important for proper vasculature remodeling during hair cycle and for timed activation of hair follicle stem cells

The precise remodeling of skin vasculature during adult hair cycle that we describe here must be tightly regulated genetically, but the factors at play are currently unknown. We find that endothelial knockout of *Alk1* induced at catagen (but importantly not when induced a few days later, at early anagen) had remarkable effects on skin vasculature organization. *Alk1* (*Activin A Receptor Like Kinase 1*) is a type I receptor for transforming growth factor-β (TGF-β) family proteins, and is essential for proper blood vessel development and branching in the yolk sac and in the early embryo (*Oh et al., 2000*). Intriguingly, it has been proposed that *Alk1* loss prolongs the activation phase and delays the resolution phase of angiogenesis (*Oh et al., 2000*) suggesting a possible correlation with our hair cycle data (see more below in this section). Furthermore, endothelial cells isolated from *Alk1*-mutant skin were shown to form irregular, and disorganized vascular structures in vitro when compared with normal cells (*Choi et al., 2013*). In telogen/early anagen, we show that proliferation of normal skin vasculature is rare and occurs mostly in the zone underneath the hair germ (HPuHG), where vasculature appears increased in the *Alk1* EndKO (these data). Given our analysis of vascular remodeling during hair cycle, it is presumable that the HPuHG is in the activation phase of angiogenesis during telogen/early anagen, which is regulated by Alk1. This would be followed by the resolution phase at full anagen when angiogenesis occurs in skin. From the HPuHG structure, it is likely that new vasculature branches out and grows vertically to populate the skin around the growing hair follicles, accounting for the increased skin vessel density at anagen.

We show here that *Alk1* expression in vasculature is required for correct density of vasculature under the hair germ and for the effective branching and dispersal of the HPuHG during late telogen and very early anagen. When the HPuHG dispersal and branching is impaired in the *Alk1* endothelial KO, dense vasculature bundles create high BMP4 concentrations under the hair germ, which we propose may be responsible for delayed HFSC activation and delayed progression into anagen. Prolonged *Alk1* loss and skin injury can result in abnormal fusions between veins and arteries, known as arteriovenous malformation (AVM) that can lead to hemorrhages (*Choi et al., 2013*; *Park et al., 2009*). Hereditary Hemorrhagic Telangiectasia, a human disease in which *Alk1* is mutated, shows prominent vasculature defects and skin AVMs in the skin (*Park et al., 2009*). Hair growth defects have not been reported for Hemorrhagic Telangiectasia, though a simple delay in anagen onset would not be easily detectable in human, where hair follicles are mostly quiescent and anagen onset is non-synchronous in most body regions (except on the scalp where hair is mostly at anagen). Our data implicating Alk1 in skin vasculature remodeling during hair cycle may aid in understanding the pathogenesis of this disease and may provide a link with its prevalence in the skin. The function of Alk1 in controlling the remodeling of the HPuHG in hair cycle supports a role of vasculature under the HFSCs activation zone in maintaining quiescence and in halting hair cycle progression (*Figure 9*, left).

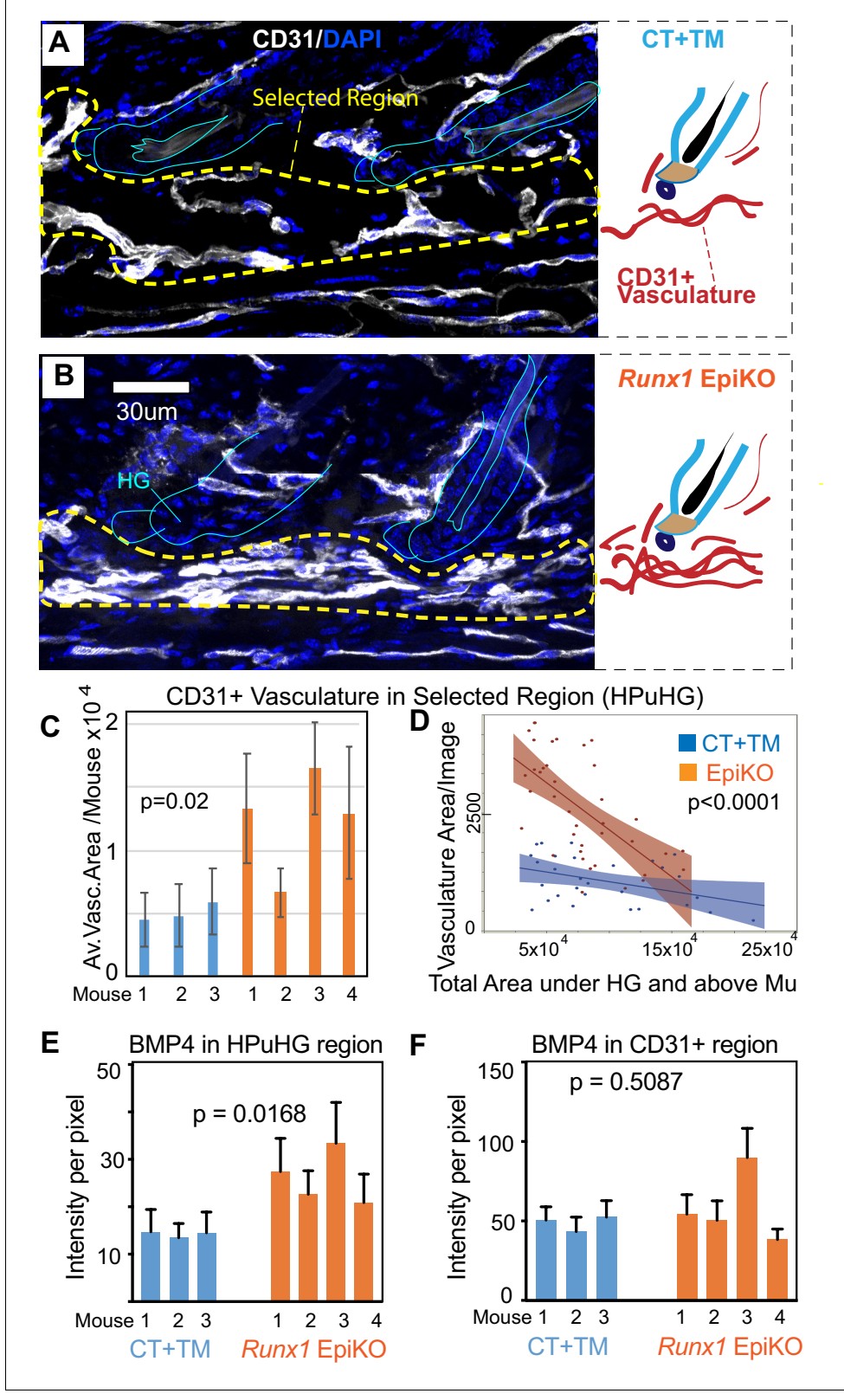

**Figure 8.** Epithelial *Runx1* knockout results in increased CD31+ vasculature and hence increased BMP4 concentration in the HPuHG region. (A–B) *Runx1*<sup>flox/flox</sup> mice with *Krt14-CreERT2* (EpiKO) or without (CT) were injected with tamoxifen (TM) at PD17, sacrificed at ~PD20. Skin sections were DAPI stained, inspected for hair morphology and CT and EpiKO samples were stage matched at catagen VIII. Images are maximum projections

*Figure 8 continued on next page*

*Figure 8 continued*

through confocal Z-stacks of 70 µm sections and show denser horizontal cables of CD31+ vasculature in the horizontal plexus under the hair germ (HPuHG) in the *Runx1* EpiKO. This is schematized in the corresponding right panels. (**C–D**) The HPuHG is quantified as the area of CD31+ vasculature in 'selected regions', as shown in A and B and *Figure 8—figure supplement 1A*, and is plotted as average among images per mouse (**C**). (**D**) Images pooled from all mice in CT and *Runx1* EpiKO groups are plotted against the total skin area under hair germ (HG) and above the muscle (Mu). See also *Figure 8—figure supplement 1A–E* for more detail and additional quantification. N = 10 images/mouse. Error bars represent standard deviation. (**E**) Quantification of average BMP4 intensity in selected HPuHG region. Same quantification method was used as in *Figure 6B*. *Runx1* EpiKO have significantly higher average intensity in the HPuHG area when compared to CT+TM. P-value was obtained using REML approach in the JMP software. N = 8–10 images/mouse. Error bars represent standard deviation. (**F**) Quantification of BMP4 intensity specifically in CD31+ HPuHG vasculature does not reveal a significant difference in signal between CT+TM and *Runx1* EpiKO. Same quantification method was used as in *Figure 6C*. P-value was obtained using REML approach. N = 8–10 images/mouse. Error bars represent standard deviation.

DOI: https://doi.org/10.7554/eLife.45977.021

The following source data and figure supplement are available for figure 8:

**Source data 1.** Spreadsheet of original quantification on Runx1 EpiKO skin (*Figure 8C and D*).
DOI: https://doi.org/10.7554/eLife.45977.023
**Source data 2.** Spreadsheet of original quantification of BMP4 in Runx1 EpiKO skin (for *Figure 8E and F*).
DOI: https://doi.org/10.7554/eLife.45977.024
**Figure supplement 1.** *Runx1* EpiKO skin shows increased vasculature in horizontal plexus under hair germ (HPuHG) at late catagen/telogen and normal vasculature at anagen.
DOI: https://doi.org/10.7554/eLife.45977.022

## *Runx1* expression in the quiescent epithelial hair follicle stem cells is important for organization of neighboring vasculature in hair cycle

Our previous work has shown that Runx1 has pleiotropic cell-autonomous effects on quiescent HFSCs including control of cell cycle factors (*Lee et al., 2013*), lipid metabolism (*Jain et al., 2018*), and Wnt signaling (*Osorio et al., 2011*). These multiple Runx1 functions prime the hair germ cells for proper timing of activation in the subsequent stage of the hair cycle, but are not sufficient for HFSC activation, as we showed previously (*Lee et al., 2014*). Here we show that *Runx1* expression in

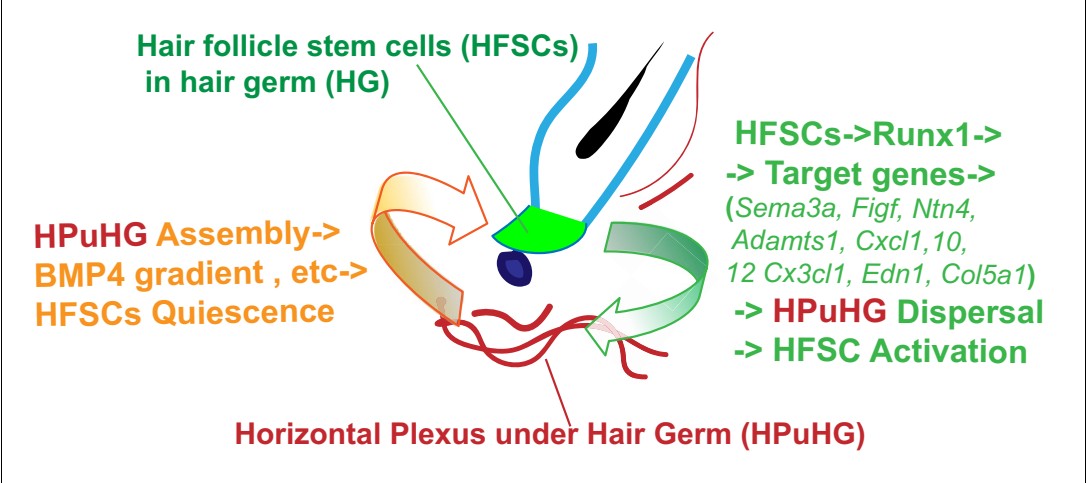

**Figure 9.** Hypothetical model of HFSCs and vasculature cross-talking in hair cycle. Left side: During late stages of HFSC quiescence (late catagen/telogen) remodeling of skin vasculature results in assembly of the horizontal plexus under hair germ HPuHG. This results in high concentrations of BMP4, a HFSC-quiescence inducing factor, near the hair germ, which may enforce telogen maintenance. Right side: HFSCs in the hair germ produce Runx1, a factor that promotes their activation. Through its target genes encoding secreted proteins with know function in vasculature remodeling, Runx1 shapes the microenvironment at late catagen/telogen to produce a less dense HPuHG. This in turn would be more permissive to HFSC activation and progression from telogen to anagen.
DOI: https://doi.org/10.7554/eLife.45977.025

the hair epithelium at catagen VIII/telogen is essential for the proper reorganization of vasculature under the hair germ. In the absence of *Runx1*, in the hair follicle, the epithelium remodels properly in quiescence prior to anagen onset, but the HPuHG is denser, showing agglomerated vasculature cables underneath the HFSC activation zone. Based on this we propose a model for future investigation in which HFSCs utilize the transcription factor Runx1 to counteract the inhibitory effect of dense vasculature array organized as horizontal plexus under the hair germ (HPuHG). We show that in turn this ensures proper BMP concentration (and likely other inhibitory signals) from the vasculature for subsequent timely SC activation.

Although more experiments are needed to understand how exactly is *Runx1* expression in the epithelium guiding the vasculature remodeling in the endothelium during the late quiescence stages of hair cycle, our previous genome-wide gene expression profiling data suggest relevant pathways for future investigation (*Lee et al., 2014*). Specifically, a handful of Runx1 target genes are secreted molecules with known function in vasculature remodeling (*Lee et al., 2014* and *Table 2* and *Figure 9*), an early observation which in fact prompted us to begin this entire study. Interfering with their function in the epithelium would help answer directly which of these molecules or combination of molecules are critical for cross-talking of epithelium with endothelium. The *Runx1*-induced changes in gene expression occur very rapidly, within 1 day of *Runx1* induction in the epithelium. In normal skin Runx1 is highly expressed in the shrinking hair bulb epithelium, as well as in the precursors of the hair germ cells (*Lee et al., 2014*). The Runx1-driven molecules secreted from the epithelium likely produce specific patterns of the extracellular environment around the shrinking hair follicle during late catagen. This would promote specific cues for migration and survival of the endothelial cells at this stage, thus resulting in the characteristic pattern of vasculature near the hair germ, with the correct density of the 'HPuHG' (*Figure 9*, right).

An interesting category of our newly uncovered Runx1-target genes are the *semaphorins*, which can signal to endothelial cells for proper cell migration and survival (*Gu and Giraudo, 2013*). Other examples of Runx1-target genes that may be potentially vasculature-relevant are extracellular matrix molecules such as *fibronectin* (*Lee et al., 2014*), a gene with known functions in vasculature development and cancer (*Kostourou and Papalazarou, 2014*), or *Colagen 5a1* implicated in familial Ehlers-Danlos syndrome with lethal arterial events (*Monroe et al., 2015*). Furthermore, several Cxcl chemokine molecules appeared upregulated in response to *Runx1* induction in the HFSCs, and chemokine signaling from neighboring cells have demonstrated effects on movement of endothelial cells and vasculature patterning (*Bosisio et al., 2014*). It is likely that synergistic effects from many of these relevant Runx1 target genes, create the complex skin vasculature patterning around the hair germ that we describe here, thus we surmise that a simple knockout of any one of these molecules may be insufficient to recapitulate the Runx1 effect on vasculature. It is worth noting that Runx1 has been previously reported to work in eye and cancer angiogenesis, cell-autonomously in the endothelial cells (*Iwatsuki et al., 2005*; *Lam et al., 2017*). To our knowledge, this is the first example of a non-cell autonomous role of Runx1, from within the SCs, to promote patterns of remodeling in the neighboring vasculature. Our data on Runx1 taken together, support a model in which HFSCs control the organization of their microenvironment, in particular the density of neighboring vasculature near the HFSC activation zone, for proper subsequent SC activation and tissue homeostasis.

In conclusion, our work provides preliminary evidence and a working model for two-directional communication and reciprocal remodeling influence between vasculature and epithelial HFSCs during the late quiescence stages of adult hair cycle. This coordination can be perturbed from each compartment by specific targeted mutation and in both cases this appears to influence subsequent timing of HFSCs activation and proper skin homeostasis. Our work lacks direct functional evidence for the molecular mechanisms that mediate the cross-talking between HFSCs and skin vasculature, but provides a handful of compelling candidates and a detailed understanding of the system for direct future investigation. Elucidating the molecular pathways at play is pending genetic targeting of the BMP4 and other candidates in the endothelium and also targeting of the many Runx1 target genes with known vasculature remodeling function in the epithelium at the critical stage of hair cycle we reveal here. Our work opens a new road of investigation in skin stem cell biology and may aid to our understanding of skin vasculature disease.

# Materials and methods

## Key resources table

| Reagent type (species) or resource | Designation | Source or reference | Identifiers | Additional information |
|---|---|---|---|---|
| Genetic reagent (*M. musculus*) | *Krt14-CreERT2* | PMID: 11034212 | RRID: MGI:2177426 | |
| Genetic reagent (*M. musculus*) | *Cdh5-CreERT2* | PMID: 20445537 | RRID: MGI:3848982 | Dr. Anne Eichmann (Yale University) |
| Genetic reagent (*M. musculus*) | *Alk1^flox^* | PMID: 17911384 | RRID: MGI:4398901 | Dr. Anne Eichmann (Yale University) |
| Genetic reagent (*M. musculus*) | *Runx1^flox^* | PMID: 15784726 | RRID: MGI:3043614 | |
| Genetic reagent (*M. musculus*) | *Rosa26-tdTomato* | Jackson Laboratory | RRID: MGI:3809523 | |
| Genetic reagent (*M. musculus*) | *C57BL/6* | Jackson Laboratory | RRID:MGI:3715241 | |
| Antibody | Anti-mouse-CD31 (Purified rat monoclonal) | BD Biosciences | BD Cat#: 550274, RRID: AB_393571 | 1:100 |
| Antibody | Anti-Ki67 (Rabbit polyclonal) | Abcam | Abcam Cat# ab15580, RRID:AB_443209 | 1:1000 |
| Antibody | Anti-Caspase-3 (Rabbit polyclonal) | R and D Systems | R and D Cat# AF835, RRID:AB_2243952 | 1:300 |
| Antibody | Anti-neurofilament-1 (Chicken polyclonal) | Millipore | Millipore Cat# AB5539, RRID:AB_11212161 | 1:1000 |
| Antibody | Anti-BMP4 (Rabbit polyclonal) | Abcam | Abcam Cat# ab39973, RRID:AB_2063523 | 1:1000 |
| Antibody | Anti-EphrinB2 (Rabbit monoclonal) | Abcam | Abcam Cat #: ab201512, RRID:AB_2810831 | 1:500 |
| Antibody | Anti-P-Cadherin (Rat monoclonal) | Thermo Fisher Scientific | Thermo Fisher Scientific Cat# 13-2000Z, RRID:AB_2533006 | 1:200 |
| Chemical compound, drug | Tamoxifen | Millipore | Millipore Cat#: T5648 | |
| Software, algorithm | ImageJ | ImageJ | RRID:SCR_003070 | |
| Software, algorithm | JMP Pro | SAS | RRID: SCR_014242 | |

## Animal studies

Mouse work followed the Cornell University Institutional Animal Care and Use Committee guidelines. For hair cycle studies, we used pure C57BL/6 wild type mice. For *Runx1* knockout (KO) studies, we used previously described mice as either *Krt14-Cre* x *Runx1^flox/flox^* (*Osorio et al., 2008*) or *Krt14-CreERT2* x *Runx1^flox/flox^* (*Scheitz et al., 2012*), as specified in text and figure legends. *Nrp1^flox/flox^* x *Cdh5-CreERT2* and *Alk1^flox/flox^* x *Cdh5-CreERT2* mouse lines have been imported in our laboratory from Dr. Anne Eichmann Yale University; individual mutations used in these crossings were previously described (*Gu et al., 2003*; *Oh et al., 2000*; *Wang et al., 2010*). For *Runx1* deletion, mice

were injected at PD17 intraperitoneally with 225 µg Tamoxifen (Sigma) per gram of body weight dissolved in corn oil and sacrificed at PD19/PD20. *Runx1^flox/flox^*+TM and *Krt14-CreERT2; Runx1^flox/flox^*+-oil littermates served as control. For *Alk1* and *Nrp1* deletion in endothelial cells, mice were injected at PD17 intraperitoneally with 200 µg Tamoxifen (Sigma) per gram of body weight dissolved in corn oil and sacrificed at PD22-PD35. *Alk1^flox/flox^*+TM and *Nrp1^flox/flox^*+TM littermates served as control. The *Rosa26-tdTomato* mice were imported from the Jackson laboratory (*Madisen et al., 2010*). The doxy-inducible *Runx1* TG mice were previously described (*Lee et al., 2014*).

## Immunofluorescence staining

Murine back skins were embedded in Optimal Cutting Temperature (OCT) compound (Tissue Tek, Sakura) on dry ice and stored at −80℃. Frozen 10–12 µm skin sections were fixed with freshly made 4% paraformaldehyde for 10 min at room temperature (RT), blocked in normal serum for 1–2 hr, and incubated with primary antibodies for 12–24 hr at 4℃, as described before (*Lee et al., 2014*; *Scheitz et al., 2012*). MOM Basic kit (Vector Laboratories) was used for mouse primary antibodies. After washing, we incubated for 1 hr with TxR, FITC, and Alexa-594 -conjugated secondary antibodies (Jackson Immuno Research). Primary antibodies include CD31 (BD Biosciences, 5502741); Ki67 (Abcam; 15580); NF-1 (chicken, EMD Millipore AB5539); BMP4 (rabbit, Abcam ab39973); EphrinB2 (rabbit, Abcam ab201512); P-Cadherin (rat, Thermo Fisher 13-2000z). Runx1 antibody was a gift from Dr. Thomas Jessell (Columbia University). For confocal imaging, 60 µm - 70 µm frozen skin sections were used with up to 48 hr antibody incubation in the cold. Immunostaining was performed as described above.

## Western blots

The skin was snap frozen in liquid N2 and the skin was ground using mortar and pestle in liquid N2. Then protein was lysed using RIPA buffer (10 mM Tris-Cl (pH 8.0), 1 mM EDTA, 1% Triton X-100, 0.1% sodium deoxycholate, 0.1% SDS, 140 mM NaCl, 1 mM PMSF, protease inhibitors). BMP4 (abcam, ab39973, 1:1000) and b-actin (Millipore Sigma, MAB1501, 1:5000) antibodies were used to detect BMP4 expression. ImageJ (http://imagej.nih.gov/ij/) was used for normalization and quantifications.

## Microscopy and image quantification

Widefield fluorescence of optical Z-stack images were collected from 60 to 70 µm skin sections stained on slides using a Nikon light fluorescence microscope (MVI) equipped with a CCD 12-bit digital camera (Retiga EXi, QImaging) and a motorized z-stage, using a 10x lens and z-step at an interval of 5 µm. To eliminate the out of focus blur, we deconvoluted z-stacks using AutoQuant X software (MVI). Single images and projections through stacks were assembled and enhanced for brightness, contrast and levels using Adobe Photoshop and Illustrator. Widefield fluorescence images of thin section (12 µm) were taken using the same equipment without using z-stack function and deconvolution.

Immunofluorescence images were taken using Zeiss LSM880 Confocal/Multiphoton Inverted Microscope ('i880') with the pinhole set to 1AU. Optical Z-stacks spaced at 0.45 to 2 µm were collected. The confocal images were analyzed and quantified using Zen Black software (Carl Zeiss) or Image J (http://imagej.nih.gov/ij/).

'CD31+ vasculature area', 'selected region', and 'total area' were defined in FIJI software as shown in *Figure 5—figure supplement 1A* and *Figure 8—figure supplement 1A*, and then quantified. Vasculature within each region of interest were selected based on CD31 signal using Freehand selections tool and area was quantified. In order to define a comparable 'selected region' of interest underneath the hair in each image, the vertical length of this area is measured in pixels. Then a freehand line was drawn underneath the hair germs on images stained for CD31 and DAPI. Using the FIJI Macros function, the line was moved downwards for a pre-measured vertical length, and the two lines were connected to form an enclosed region of interest. We found that this method encompasses fully the horizontal plexus underneath the hair germ in wildtype and in *Runx1* KO images, while keeping the 'selected region' comparable in size between images. However, in *Alk1* EndKO skin the HPuHG characteristic structure exceeded the selected region and was defined qualitatively by freehand drawing for data in *Figure 5A,B*, whereas a fixed area was kept for quantification in

*Figure 5—figure supplement 1*. For quantification of both *Alk1* and *Runx1* mutant mice, the 'total area' between hair germ and muscle encompassing hypodermis were also measured using a free-hand line outlining the border with the muscle area. The muscle structure was identified in images prior to contrasting and channel splitting, as shown by characteristic background fluorescence pattern in this structure of the skin subcutis.

BMP4 intensity in HPuHG was measured by selecting the HPuHG region, and the average intensity per pixel was calculated. Three rectangular boxes were randomly drawn in the background, and the average intensity per pixel was measured to obtain background intensity. The final HPuHG intensity was calculated as HPuHG raw intensity minus background intensity. For BMP4 intensity in CD31 + area in HPuHG region, CD31+ area was outlined from the HPuHG region using free hand selection tool, and the average raw intensity of BMP4 in those selected areas was measured in ImageJ. Then three rectangular areas in the background were randomly selected and average intensity per pixel was measured as background intensity. The final BMP4 intensity in CD31+ area was determined as raw intensity minus average background intensity. For BMP4 intensity in IFE, IFE regions were outlined using the freehand selection tool, and the average raw intensity of BMP4 per pixel in those regions was measured. The final BMP4 intensity in IFE was obtained by raw IFE BMP4 intensity minus average background intensity.

Quantification of C57BL/6 wild type total skin CD31+ vasculature during hair cycle was done by selecting vessels in the entire skin area above the muscle layer. The HPuHG area was defined as described above for the *Runx1* mutant mice in a selected region underneath the hair germ. Angles of vasculatures were obtained by drawing a line along the vasculature branches in total skin and measuring its angle relative to a horizontal line parallel to the epidermis.

Representative images were enhanced for better quality using Brightness/Contrast tool in Adobe Photoshop and Illustrator. Autofluorescence from paper towel was removed in *Figure 4C*, *Figure 1—figure supplement 1E–G*, and *Figure 4—figure supplement 1C*.

## Statistical analysis

Statistical analysis was performed using JMP Pro 14. For statistical analysis performed on vasculature data from *Runx1* and *Alk1* mutant mice, genetic background, total area, and individual variance were considered as random effects attributed to individual mouse. Area of vasculature in our selected region of interest was our variable and p-values were calculated by two-way ANOVA and Restricted maximum likelihood (REML) analysis with unbounded variance components. P-values of statistical analysis related to BMP4 and Pcad$^{High}$ cell counting were calculated by REML method mentioned above using each individual mouse as the random effect. For statistical analysis performed for vasculature changes during hair cycle analysis in wild type BL6 mice p-value was calculated from Student's t-test. For differences between the distribution of vasculature angles in each hair cycle stage, p-values were calculated by Kolmogorov-Smirnov two-sample test.

Analyses for *Figure 7* used JMP analysis on the data of vasculature length of contact with follicles, which were stage matched at telogen. Data were log transformed and separated by mouse ID to account for individual mouse variation. REML analysis was run on JMP, with mouse ID set as a random effect, location (germ, lower bulge, upper bulge) and fixed conditions (mutant type and its control). The log transformed values for length of contact were measured on the Y-axis and then the analysis was run as least squares means plot, generated by the program and used for analysis.

## Acknowledgements

We thank Dr. Anne Eichmann from Yale University for sharing the *Nrp1$^{flox/flox}$* x *Cdh5-CreERT2* and the *Alk1$^{flox/flox}$* x *Cdh5-CreERT2* mice and for useful discussion. We also thank Cornell Statistical Consulting Unit for statistical consultation. Confocal microscopy work on Zeiss i880 was supported by Cornell BRC facility (NYSTEM C029155 and NIH S10OD018516). The work in this paper was funded by NIH/NIAMS R01 AR073806 and R01 AR070157 to TT and post-doctoral NYSTEM training awarded (DOH01-C30293GG-3450000) to PJ.

## Additional information

### Funding

| Funder | Grant reference number | Author |
|---|---|---|
| National Institute of Arthritis and Musculoskeletal and Skin Diseases | RO1 AR070157 | Tudorita Tumbar |
| New York State Stem Cell Science | DOH01-C30293GG-3450000 | Prachi Jain |
| National Institute of Arthritis and Musculoskeletal and Skin Diseases | RO1 AR073806 | Tudorita Tumbar |

The funders had no role in study design, data collection and interpretation, or the decision to submit the work for publication.

### Author contributions

Kefei Nina Li, Data curation, Formal analysis, Validation, Investigation, Visualization, Methodology, Writing—review and editing; Prachi Jain, Conceptualization, Data curation, Formal analysis, Supervision, Validation, Investigation, Visualization, Methodology, Writing—review and editing; Catherine Hua He, Data curation, Formal analysis, Investigation, Methodology, Writing—review and editing; Flora Chae Eun, Data curation, Formal analysis, Investigation, Methodology; Sangjo Kang, Formal analysis, Visualization, Methodology; Tudorita Tumbar, Conceptualization, Resources, Data curation, Formal analysis, Supervision, Funding acquisition, Validation, Investigation, Visualization, Writing—original draft, Project administration

### Author ORCIDs

Kefei Nina Li ![ORCID] https://orcid.org/0000-0002-4361-629X
Prachi Jain ![ORCID] https://orcid.org/0000-0002-1657-7045
Tudorita Tumbar ![ORCID] https://orcid.org/0000-0002-2273-1889

### Ethics

Animal experimentation: Mouse work followed the Cornell University Institutional Animal Care and Use Committee guidelines. IACUC protocol # 2007-0125.

### Decision letter and Author response

Decision letter https://doi.org/10.7554/eLife.45977.028
Author response https://doi.org/10.7554/eLife.45977.029

## Additional files

### Supplementary files

• Transparent reporting form
DOI: https://doi.org/10.7554/eLife.45977.026

### Data availability

All data generated or analysed during this study are included in the manuscript and supporting files.

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
