## [Decision Letter]

Thank you for submitting your article "Skin vasculature and hair follicle stem cells cross-talking associated with stem cell activation and tissue homeostasis" for consideration by *eLife*. Your article has been reviewed by three peer reviewers, one of whom is a member of our Board of Reviewing Editors, and the evaluation has been overseen by Marianne Bronner as the Senior Editor. The following individual involved in review of your submission has agreed to reveal their identity: Xing Dai (Reviewer #3).

The reviewers have discussed the reviews with one another and the Reviewing Editor has drafted this decision to help you prepare a revised submission.

Summary:

This manuscript addresses the role of crosstalk between blood vessels and hair follicles in the skin. The authors demonstrate that changes in vessel remodeling occur during the hair follicle cycle. The data also indicate that loss of the BMP receptor Alk1 in endothelial cells alters hair follicle cycling. Furthermore, the authors demonstrate that Runx1 depletion in the epidermis results in defects in vascular remodeling. This work pioneers in describing the vasculature organization of the skin endothelium during different stages of the hair cycle, alluding to the importance of this organization. Moreover, the authors have identified the precise time-point at which vasculature remodeling around the hair germ is essential for proper hair cycle progression. Altogether, the work presented in this article will be of interest to the broad audience of *eLife* and will significantly advance fields of skin stem cell biology and stem cell-niche interactions.

Essential revisions:

1) Expression of BMP4 by endothelial cells.

A) The upregulation of BMP4 in endothelial cells from Alk1 EndKO mice is a critical point. Since BMP4 is also expressed by other cell types (e.g., IFE cells in control skin), quantitative analysis of signal intensity in HPuHG/vasculature and IFE of control and KO skin would be important to clarify points such as whether Alk1 EndKO-induced BMP4 expression change is unique to the endothelial cells, and whether this is an increase in the number of expressing cells or increase in the expression level in individual cells. Additionally, sorting of CD31+ endothelial cells followed by expression analysis for Bmp4 would be informative although not essential.

B) Is there evidence that BMP signaling is elevated in hair follicles from Alk1 endoKO mice?

C) Does Runx1 deletion alter BMP expression in the vasculature?

2) Analysis of endothelial morphology in mutant mice.

Figure 3—figure supplement 2: First, the authors need to show the same immunofluorescence (IF) experiment (CD31, Ephrinb2, NF1 and/or Ki67) carried out in both CDH5-CreERT2 x Nrp1 fl/fl TM PD17 and CDH5-CreERT2 x Alk1 fl/fl TM PD17 at a time point where the phenotypic differences in Alk1EndKO are observed.

Also, the Nrp1f/f TM PD17 mice should also be sacrificed at PD31 to show Ki67 staining in this skin is comparable to CT. Additionally, did the authors induce with TM at PD21 for CDH5-CreERT2 x Nrp1 fl/fl mice? The inclusion of this treatment will eliminate the contribution of Nrp1 during the hair cycle.

3) Additional hair cycle analysis in mutant mice

A) The authors should include a panel of treated CDH5-CreERT2 x Alk1 fl/fl TM PD17 mice sacrificed at early/mid Telogen (PD 20/25). Does loss of Alk1 affect progression through telogen, or it merely delays entry into anagen? This data should be included to address at what time point the hair cycle is delayed in the absence of Alk1.

B) Figure 4—figure supplement 3: The authors show that the Ki67 staining signature to be vastly different in PD28/31 skin of CT+TM when compared to Alk1 EndKO. The morphology of the hair follicle (HF) was also different as shown in the representative images of the two skins (Figure 3C). The authors need to provide data showing that at PD28 there are no significant differences in hair cycle between CT+TM and EndKO skins.

C) Greco et al., 2009, have previously shown that Pcad^High^ cells are maintained during telogen in the hair germ (HG). Are the number of cells in the hair germ maintained during telogen in Alk1EndKO? Or is the delay a result of the loss of HG cells? The inclusion of this data will ascertain the conclusion that the delay in the hair cycle is attributed to only vasculature organization and not the loss of HG cells.

4) Edit some of your conclusions based on the data presented.

While data in this work support a tight association between the indicated vasculature changes (assembly vs. dispersion of HPuHG) and HFSC quiescence and do suggest promising leads for future investigation, definitive evidence for a causal relationship is still lacking. The involvement of the skin vasculature in general (as opposed to their specific pattern of distribution and spatial relationship with the HFSCs creating a signaling gradient) cannot be fully excluded. As such, several statements in the manuscript can be tuned down a bit. Examples include "Impact statement: Hair follicle stem cells priming and skin vasculature remodeling are coordinated during quiescence to promote the proper timing of stem cell activation and initiation of a new hair growth cycle", and "These data suggest that increased vasculature near the HFSC activation zone is inhibitory to their activation and delays progression into the hair growth cycle".

5) More quantification of data.

Quantification, as is nicely done in some figures, needs to be done in others (e.g., Figure 1 – the amount and localization of the vasculature and number of mice/sections analyzed; Figure 3B – the number of CD31/tdT positive cells).

---

## [Author Response]

Essential revisions:1) Expression of BMP4 by endothelial cells.A) The upregulation of BMP4 in endothelial cells from Alk1 EndKO mice is a critical point. Since BMP4 is also expressed by other cell types (e.g., IFE cells in control skin), quantitative analysis of signal intensity in HPuHG/vasculature and IFE of control and KO skin would be important to clarify points such as whether Alk1 EndKO-induced BMP4 expression change is unique to the endothelial cells, and whether this is an increase in the number of expressing cells or increase in the expression level in individual cells. Additionally, sorting of CD31+ endothelial cells followed by expression analysis for Bmp4 would be informative although not essential.

We tried to sort CD31+ cells, but our cell isolation methods using trypsin and collagenase destroy the antigen, so we were unable to get the staining to work in these cells. However, we quantified the BMP4 intensity in the HPuHG area, using immunostaining of CD31 and BMP4 on skin sections. We used 12μm skin sections from mice that were stage-matched at the Telogen to Anagen transition (when rare Ki67+ cells are present in the hair germ), using *Alk1* CT and KO (n=3 per group). 9-16 images per mouse were taken under the same exposure (250ms for BMP4). In ImageJ, the HPuHG region was selected, and the average intensity per pixel was calculated. Three rectangular boxes were randomly drawn in the background region, and the average intensity per pixel was measured to obtain background intensity. The final HPuHG intensity was calculated as HPuHG raw intensity/pixel minus background intensity/pixel. Statistics was performed using REML approach, which took into consideration mouse to mouse variation in comparing CT and KO groups. The calculated p-value is 0.0069, indicating the significant increase in BMP4 concentration in HPuHG area in *Alk1* KO compared to CT. The quantification is now shown in new Figure 6B.

BMP4 intensity in endothelial cells per se, was also measured using the same images that were used for above analysis. Specifically, all the CD31+ pixels that are in the HPuHG region or partly within the HPuHG region were selected in ImageJ. Then BMP4 intensity per pixel in those CD31+ region was measured. Again, background values that were calculated before were subtracted from the raw BMP4 intensity in CD31+ region to get the final BMP4 intensity per pixel in CD31+ region. Statistics was performed using REML approach, and the calculated p-value is 0.2103. These results indicated there is no change in BMP4 expression by endothelial cells between *Alk1* CT and KO. Since we showed that there is an increase in the CD31+ area in the HPuHG region, it is reasonable to infer that the BMP4 increase comes from the increased vasculature density in this area. The quantification is shown in new Figure 6C. With that said, some BMP+ cable vascular-like structures in the HPuHG region of the *Alk1* KO have no, or perhaps low, CD31 and their identity will remain to be investigated in more details later.

BMP4 intensity in IFE was also analyzed, as requested by reviewers. For this analysis, 8-10 images per mouse were used. IFE regions were hand-selected in each image in the ImageJ software, and the BMP4 intensity per pixel was measured. After subtracting background intensity as described above for measurement in HPuHG, we got the final IFE BMP4 intensity. Statistics was performed using REML approach, and the calculated p-value is 0.9321. These results indicated there is no change in BMP4 expression by IFE keratinocytes in *Alk1* EndoKO mice. The quantification is shown in new Figure 6D.

Overall, our quantitative analysis indicates there is an increase in BMP4 local concentration near the hair germ (in HPuHG). This increase is not caused by increased expression of BMP4 by endothelial cells or keratinocytes, and can be attributed to increased CD31+ vasculature density in the HPuHG.

B) Is there evidence that BMP signaling is elevated in hair follicles from Alk1 endoKO mice?

To address this question, 12μm *Alk1* KO and CT skin sections were immunostained with phospho-SMAD 1/5/8 (pSMAD), as it is an indication of BMP signaling. Once BMP4 binds to its receptor, then SMAD 1/5/8 will be phosphorylated and translocated to the nucleus. We observed nuclear pSMAD staining in a fraction of the hair follicles in PD31 (Telogen-Anagen II) and PD35 (Anagen I-II) *Alk1* EndoKO, indicating the persistence of BMP4 signaling (data not shown). Unfortunately, the intensity of the P-Smad staining was highly dependent on the age of the blocks, with the older blocks not giving any signal at all, and the newer blocks staining more strongly. Because of this technical problem and after a lot of staining troubleshooting, and given the difficulty of obtaining sufficient new knockout mice in a timely manner, we decided is best to hold back on these data and combine it with future experiments where we will explore in more depth the relevance of BMP signaling to the cross-talking between vasculature and hair follicle.

C) Does Runx1 deletion alter BMP expression in the vasculature?

Immunostaining of CD31 and BMP4 was performed on 12μm staged-matched *Runx1* PD17-20 CT and KO (Catagen VIII) skin sections. 5-8 images per mouse (n=3 per group) were taken under the same exposure (250ms for BMP4). BMP4 expression in CD31+ vasculature was quantified using the same method as for *Alk1* KO skin. Statistics results showed a p-value of 0.5087. This result showed that *Runx1* deletion did not alter BMP4 expression in the vasculature itself, meaning endothelial cells express similar amounts of BMP.

Since we found *Runx1* mutant also has increased vasculature density in the HPuHG region, we also tested whether local BMP4 level in this region was increased (as it was in the *Alk1* EndKO). Thus, we quantified BMP4 intensity in HPuHG region using the same method as we did for *Alk1* mutant, and statistics showed a p-value of 0.0168. This result further supported our conclusion that increased HPuHG vasculature leads to increased BMP4 local concentration, and this true in the *Runx1* EpiKO mice as well. These data are shown in new Figure 8E and F.

2) Analysis of endothelial morphology in mutant mice.Figure 3—figure supplement 2: First, the authors need to show the same immunofluorescence (IF) experiment (CD31, Ephrinb2, NF1 and/or Ki67) carried out in both CDH5-CreERT2 x Nrp1 fl/fl TM PD17 and CDH5-CreERT2 x Alk1 fl/fl TM PD17 at a time point where the phenotypic differences in Alk1EndKO are observed.

We now provide additional evidence for a vasculature phenotype in the *Alk1* Endo KO in 2 groups of mice that were stage matched at either telogen (group 1) or at telogen/anagen1 (group 2), when rare Ki67+ cells are present in the hair germ – now shown in new Figure 5B.

Also, the Nrp1f/f TM PD17 mice should also be sacrificed at PD31 to show Ki67 staining in this skin is comparable to CT. Additionally, did the authors induce with TM at PD21 for CDH5-CreERT2 x Nrp1 fl/fl mice? The inclusion of this treatment will eliminate the contribution of Nrp1 during the hair cycle.

Unfortunately, we are unable to perform any new *Nrp1* experiments because we no longer have the *Nrp1* mutant mice. We did the initial screen quite a few years ago when we began this project, and because we did not see any phenotype in *Nrp1* KO mice whereas the *Alk1* Endo KO mice showed a robust and consistent phenotype both in hair cycle and in vasculature remodeling, we did not maintain the *Nrp1* mouse line and focused on *Alk1* mice instead. In our previous submission, we reasoned it’d be valuable for the community to provide the *Nrp1* data, even though was negative. However, we agree that the lack of *Nrp1* involvement in vasculature remodeling during the hair cycle may not be convincing enough based on existing data, thus we decided to remove this part from our manuscript since it does not add to our story.

3) Additional hair cycle analysis in mutant miceA) The authors should include a panel of treated CDH5-CreERT2 x Alk1 fl/fl TM PD17 mice sacrificed at early/mid Telogen (PD 20/25). Does loss of Alk1 affect progression through telogen, or it merely delays entry into anagen? This data should be included to address at what time point the hair cycle is delayed in the absence of Alk1.

We produced as many additional animals as we could for this revision in the past 4 months, and added a number of earlier time points to the analysis. We now have analyzed a total of 24 CT and 15 KO mice, on which we did immunostaining of Ki67 on 12μm skin sections to precisely establish the hair cycle stages at different ages. These mice are now summarized in Table 1 and in Figure 4E. Based on these results, we firmly conclude that KO mice enter telogen, but do not progress to anagen in a timely manner, due to a delay in stem cell activation in the hair germ.

B) Figure 4—figure supplement 3: The authors show that the Ki67 staining signature to be vastly different in PD28/31 skin of CT+TM when compared to Alk1 EndKO. The morphology of the hair follicle (HF) was also different as shown in the representative images of the two skins (Figure 3C). The authors need to provide data showing that at PD28 there are no significant differences in hair cycle between CT+TM and EndKO skins.

We apologize about this confusion. The intention was to use mice at matched hair cycle stages, and at PD28/31 where we examined a total of 18 mice we found enough CT and KO mice where follicle were at the same stage (Tel/Ana I – with rare KI67+ in the germ) and these stage-matched mice were used for quantification of vasculature. This was confusing because the majority of CT mice were at Ana II+, as indicated by our KI67 staining and shown in Figure 4C and E, and Figure 4—figure supplement 1C. In addition to mice stage matched at telogen-anagen I transition, we have now also used younger mice, stage matched at telogen. The quantification of both stages is shown in new Figure 5B. As you can see, both at telogen and at Telogen/Anagen I End KO skin shows abnormal increase in density of the HPuHG, at a level never observed in any of the CT mice at any of the stages analyzed. In addition, the CT HPuHG density decreased at the transition from telogen to anagen I in (as also show in our analysis of vasculature in hair cycle in BL6 WT mice, Figure 1 and 2). However, in the *Alk1* EndKO the HPuHG remains strongly packed and does not disperse at the transition between the telogen and anagen I. With that said, skin from *Alk1* EndKO found at Anagen I/II shows that dispersal of the HPuHG eventually occurs, although the vast increase in CD31 signal throughout the skin persists into this stage.

C) Greco et al., 2009, have previously shown that Pcad^High^ cells are maintained during telogen in the hair germ (HG). Are the number of cells in the hair germ maintained during telogen in Alk1EndKO? Or is the delay a result of the loss of HG cells? The inclusion of this data will ascertain the conclusion that the delay in the hair cycle is attributed to only vasculature organization and not the loss of HG cells.

12μm stage-matched telogen *Alk1* CT and KO skin sections were immunostained with Pcad and Pcad^High^ cell number was counted. 10-20 hair follicles per mouse were analyzed. Statistics was performed using REML approach, and had a p-value of 0.2970 suggesting that the hair germ cells were indeed maintained in the *Alk1* Endo KO skin. This result ascertains the conclusion that the delay in hair cycle is not caused by loss of Pcad^High^ hair germ cells. The data are shown in new Figure 4—figure supplement 1B.

4) Edit some of your conclusions based on the data presented.While data in this work support a tight association between the indicated vasculature changes (assembly vs. dispersion of HPuHG) and HFSC quiescence and do suggest promising leads for future investigation, definitive evidence for a causal relationship is still lacking. The involvement of the skin vasculature in general (as opposed to their specific pattern of distribution and spatial relationship with the HFSCs creating a signaling gradient) cannot be fully excluded. As such, several statements in the manuscript can be tuned down a bit. Examples include "Impact statement: Hair follicle stem cells priming and skin vasculature remodeling are coordinated during quiescence to promote the proper timing of stem cell activation and initiation of a new hair growth cycle", and "These data suggest that increased vasculature near the HFSC activation zone is inhibitory to their activation and delays progression into the hair growth cycle".

We agree that more evidence is needed to draw firm conclusion and that is why we use words such as ‘suggest’ rather than ‘demonstrate’, and so on. We want to give space for alternative explanation but also make it clear what our model is; in the new text we tried harder to strike the right balance in discussing our results, as requested by reviewers.

5) More quantification of data.Quantification, as is nicely done in some figures, needs to be done in others (e.g., Figure 1 – the amount and localization of the vasculature and number of mice/sections analyzed.

We quantified the total vasculature in PD19, PD20, PD21 and PD28 BL6 mice, and we added a quantification of vasculature in HPuHG region in PD19, PD20 and PD21 WT BL6 mice. For vasculature quantification, statistics was performed using REML methods. In addition, to show the organization of the skin vasculature is changing during hair cycle in terms of orientation relative to a horizontal line parallel to the epidermis we measured the angle of vasculature in total skin relative to a horizontal line parallel to the epidermis. Kolmogorov-Smirnov two-sample test was performed to show the differences in distribution of angles. These quantifications, together with number of mice / sections analyzed constitute an entirely new figure, and are now shown in new Figure 2.

Figure 3B – the number of CD31/tdT positive cells).

tdTomato was quantified by measuring the total area of tdTomato+ CD31+ region, and divide by the total area of CD31+ region, and found that up to 30% CD31+ cells are also tdTomato+. No tdTomato+ cells were observed outside the CD31+ area. This analysis is now added to text.